# Mastering Domain Shift Image Enhancement Via Differentiable Physics

## Abstract

Visual perception in the wild have demonstrated transformative potential across a wide range of applications, spanning from planetary exploration to deep-sea monitoring missions. However, a fundamental challenge remains in enabling visual perception enhancement that can explicitly extract rules and support interactive, precise manipulation in unknown, dynamic environments—particularly under conditions of large scale data absence, heterogeneous data distribution, and without the supervision of annotated images. Our approach introduces a differentiable physics framework that unifies the camera response model (CRM) with deep learning to achieve visual perception enhancement under multiple degradation conditions. Specifically, grounded in fundamental principles of radiation physics, we formulate the camera response function (CRF) calibration as a constrained optimization problem. Then we reconstruct the brightness transformation function (BTF) in traditional CRM as a multi-scale generative network, completely decoupling it from the CRF. Meanwhile, we design a dual-branch contrastive encoder that enables the BTF to regulate the irradiance enhancement process through multi-scale exposure distributions learned from guide images. This offers a flexible BTF interface supporting stable and controllable domain generalization for image enhancement. Through comprehensive experiments, our method significantly advances domain generalization capabilities in adaptive image enhancement, outperforming specialized counterparts by margins of +1.226 (UIQM) averaged across challenging unseen underwater domains.

## 1 Introduction

Visual perception in the wild faces the fundamental challenges of data scarcity and unknown environmental dynamics. Furthermore, the degradation types in wild environments exhibit a highly heterogeneous nature, and the underlying degradation mechanisms vary significantly across different scenarios—such as intense radiation noise on planetary surfaces and aqueous scattering-induced blur in deep-sea environments. Traditional deep neural network methods Xu et al. (2020b); Shi et al. (2024); Lv et al. (2024); Wang et al. (2024a) lack physical interpretability, and their generalization capability is constrained by the coverage of training data, making them ineffective in achieving cross-domain image enhancement. For instance, the aforementioned methods Wang et al. (2024a); Shi et al. (2024), although capable of cross-domain image enhancement, are constrained by their mandatory reliance on cross domain data during training, which fundamentally limits their performance when operating without any target-domain priors. As demonstrated in **Appendix** Figure 9, when trained exclusively on single-domain low-quality datasets, both comparative models exhibit substantial performance degradation in unseen distorted domains, failing to meet expected enhancement requirements.

To enhance the adaptability of models in complex dynamic environments and improve their inherent interpretability, the fundamental solution lies in designing constraints for models based on explicit physical rules or optical models. Recent theoretical analyses Ren et al. (2018); Mao et al. (2025) suggest the CRM provides a promising pathway. CRM mathematically decomposes imaging physics into CRF and BTF, theoretically enabling radiometric calibration. However, existing BTFs rely on exposure ratio estimation rooted in classical Retinex theory, thereby only exhibiting a singular brightness adjustment capability and failing to address the issue of improving comprehensive visual quality in aspects such as chromaticity, contrast, and sharpness. Furthermore, in dynamic and open

environments, there exist significant discrepancies among degraded domains, and current BTF-based approaches cannot achieve effective generalization across different degraded domains.

To overcome the challenges aforementioned, building upon the CRM and integrating principles of radiometry, we construct a differentiable physical model for more flexible image enhancement: i) Decoupling the strong correlation between the CRF and the BTF, we reconstruct the modeling of the traditional BTF (based on simple nonlinear functions) to be implemented by an generative network. Meanwhile, a physical constraint is designed for this generative network to guide its optimization trajectory. ii) Furthermore, to enhance the domain generalization capability and flexibility of the method, we design an adaptive feature fusion scheme based on empirical cumulative distribution functions (eCDFs) matching, introducing an interactive and precise regulation mechanism for the BTF. These improvements necessitate corresponding innovations in network design and feature extraction mechanisms. Specifically, to materialize the reconstructed BTF framework (i) and the adaptive fusion scheme (ii), we develop two core technical innovations: **(1) Stable Calibration of CRF:** Guided by fundamental physical principles, we derive the data acquisition process required for CRF calibration and formulate a constrained optimization problem which is subsequently solved using the ADMM algorithm. **(2) Latent Feature Extraction:** To optimize the effect of adaptive feature fusion, we design a dual-branch contrastive encoder. This encoder employs a symmetrical sampling strategy and adheres to the principle of mutual information maximization, aiming to extract highly discriminative features, which consequently significantly enhances the generalization capability of the proposed method across diverse degradation domains.

Our contributions are fourfold: (1) We replace traditional nonlinear BTF with an image generative network by decoupling CRF-BTF correlation, enabling flexible brightness transformation. (2) We derive the fundamental theory for CRF calibration based on radiometric principles. Combined with the EMA-ADMM algorithm, more stable weighting values for the camera response basis functions are obtained. (3) We propose CAE that extracts compact intra-domain latent features with high inter-domain separability, consolidates cross-scenario/device invariant representations to preserve texture details and rectify radiometric properties, thereby enhancing cross-domain discriminability. (4) We propose an adaptive eCDFs matching mechanism to generate implicit exposure rate representations for BTF, enabling target-domain high-quality latent features to guide enhanced image generation with domain-shift adaptation.

## 2 METHODOLOGY

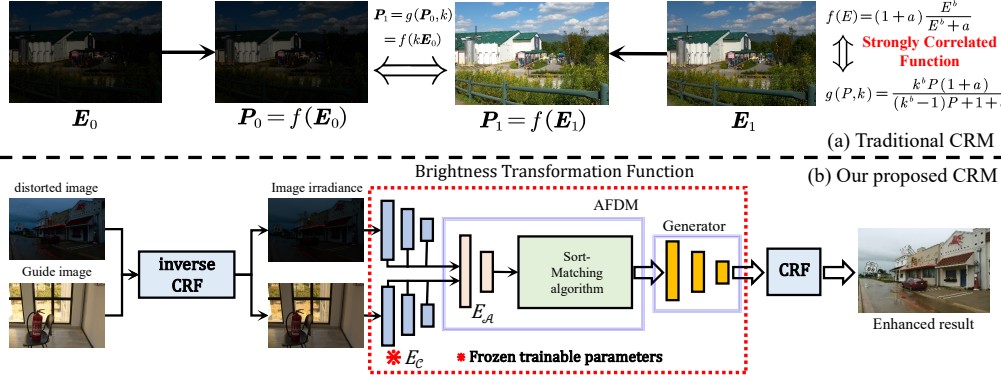

Figure 1: Comparisons of our proposed CRM and conventional CRM

### 2.1 PROBLEM FORMULATION

Computer vision algorithms often assume image intensity reflects scene irradiance, but cameras use nonlinear in-camera processes. Ignoring these degrades algorithms relying on irradiance/illumination estimation. To model such nonlinearities, CRF links scene irradiance to pixel values. When exposure changes, sensor irradiance varies linearly, but image intensity typically doesn't-creating a nonlinear BTF between differently exposed images of the same scene. CRF and

BTF form the CRM, with the comparametric equation $g(f(\bullet), k) = f(k\bullet)$ relating $f$ and $g$ for mutual conversion where $\bullet$, $k$ stand for irradiance value of any image and exposure ratio, respectively.

Given a distorted image $\mathbf{P}_x$, in traditional CRM based image enhancement algorithms, the enhancement process can be described as:

$$\widehat{\mathbf{P}}_x = f(\mathbf{E}_x \circ k) = g(f(\mathbf{E}_x), \mathbf{1} \otimes \mathbf{L}_x) = g(\mathbf{P}_x, \mathbf{1} \otimes \mathbf{L}_x), \tag{1}$$

where $\mathbf{L}_x$ stand for the illumination map of distorted image $\mathbf{P}_x$. It is calculated by Retinex decomposition ($\mathbf{P}_x = \mathbf{R}_x \circ \mathbf{L}_x$). $\mathbf{E}_x$ represents the image irradiance of $\mathbf{P}_x$.

Thus, to satisfy equation $g(f(\mathbf{E}_x), k) = f(k\mathbf{E}_x)$, $g$ is usually derived from $f$, and they are deeply coupled, resulting in poor flexibility. Meanwhile, solving the exposure ratio $k$ relies heavily on traditional Retinex theory, which is completely ineffective for domain shifts in dynamic open environments.

Addressing the practical pain points of existing CRM based image enhancement methods, our research focuses on improving the BTF to enable flexible irradiance enhancement, followed by deriving the final enhanced image pixel values via CRF. We provide a concise illustration in Figure 1. Specifically, leveraging the capability of generative adversarial networks (GANs) in image generation, improvements are first made to the equation $g(f(\mathbf{E}_x), k) = f(k\mathbf{E}_x)$. Concurrently, leveraging the advantage of contrastive learning in extracting highly discriminative features, we introduce guide images for cross-domain latent feature extraction. This approach enhances the domain shift robustness of the proposed algorithm and enables effective domain generalization.

## 2.2 CRF CALIBRATION

To obtain accurate irradiance values from pixel values, we first need to calibrate the CRF of the optical sensor. To achieve CRF calibration without relying on any sensor configuration information, we first derive the following theorem based on radiometric principles:

**Theorem 1.** *For two color patches $i$ and $j$ in the Macbeth ColorChecker under the same illumination field, we have $\frac{S_i}{S_j} = \frac{R_i \cdot I}{R_j \cdot I} = \frac{R_i}{R_j} \Rightarrow \frac{E_i}{E_j} = \frac{R_i}{R_j}$, where $S_{i,j}$ and $R_{i,j}$ denote the scene radiances and reflectances of color patches $i$ and $j$, respectively.*

**Proof:** See **Appendix** for details. Theorem 1 indicates that the ratio of image irradiances $E_i$ and $E_j$ generated by the two color patches is equal to the ratio of their reflectances $R_i$ and $R_j$. These reflectances are pre-measured with high precision and publicly available. According to the definition of CRF in Grossberg & Nayar (2003), $f$ is a monotonically increasing function satisfying $f(0) = 0$ and $f(1) = 1$. Thus, we can set the irradiance of the reference white patch to 1, and further derive the irradiances of all other arbitrary color patches.

Due to the limited camera types and outdated models in the DoRF dataset Grossberg & Nayar (2003), directly modeling the CRF of modern cameras using this dataset is infeasible. To construct a Camera Response Model (CRM) based on the CRFs of new camera models, we draw inspiration from EMoR and derive its general form as $f = f_0 + \sum_{i=0}^{M} c_i^* q_i$. Here, the basis function matrix $\mathbf{H}$ is constructed using $n$ pairs of sampling points $\left(\widehat{E}_1, \widehat{P}_1\right), \cdots, \left(\widehat{E}_n, \widehat{P}_n\right)$ $(\widehat{E}_1 < \widehat{E}_2 < \cdots < \widehat{E}_n)$ collected from the new camera:

$$\mathbf{H} = \begin{bmatrix} q_1(\widehat{E}_1) & q_2(\widehat{E}_1) & \cdots & q_M(\widehat{E}_1) \\ \vdots & \vdots & & \vdots \\ q_1(\widehat{E}_n) & q_2(\widehat{E}_n) & \cdots & q_M(\widehat{E}_n) \end{bmatrix}_{n \times M} \tag{2}$$

where $p_i$ are obtained by performing Principal Component Analysis (PCA) on 171 selected curves from the DoRF dataset Grossberg & Nayar (2003).

Subsequently, the problem of solving for $\mathbf{c}^* = [c_1^*, \ldots, c_M^*]^T$ is transformed into an optimization problem that minimizes the cost function $\|\mathbf{Hc} - \mathbf{b}\|^2$. Here, $f_0$ represents the average curve of CRFs in the DoRF dataset Grossberg & Nayar (2003), and $\mathbf{b} = [\widehat{P}_1 - f_0(\widehat{E}_1), \ldots, \widehat{P}_n - f_0(\widehat{E}_n)]^T$. To enforce the monotonicity of the CRF to be solved, we construct a discrete differential operator $D$ as

follows:

$$\Delta \widehat{E}_i = \widehat{E}_{i+1} - \widehat{E}_i \quad (i = 1, 2, \cdots, n-1), \quad D_{i,j} = \begin{cases} -\frac{1}{\Delta \widehat{E}_i} & \text{if } j = i \\ +\frac{1}{\Delta \widehat{E}_i} & \text{if } j = i+1 \\ 0 & \text{otherwise} \end{cases} \tag{3}$$

The original optimization problem is formulated as:

$$\min_{\mathbf{c}} \frac{1}{2} \|\mathbf{Hc} - \mathbf{b}\|^2$$
$$\text{s.t.} \quad [D(\mathbf{Hc} + \mathbf{f}_0)]_i \geq 0, \quad \mathbf{f}_0 = \left[ f_0\left(\widehat{E}_1\right), f_0\left(\widehat{E}_2\right), \cdots, f_0\left(\widehat{E}_n\right) \right]^{\mathrm{T}} \tag{4}$$

To derive its equivalent form using the Alternating Direction Method of Multipliers (ADMM), we first introduce an auxiliary variable $\mathbf{z}$. By rewriting the constraint with the indicator function $I_{\mathcal{K}}(\mathbf{z})$ (where $I_{\mathcal{K}}(\mathbf{z}) = 0$ if $\mathbf{z} \geq 0$, and $I_{\mathcal{K}}(\mathbf{z}) = +\infty$ otherwise; $\mathcal{K} = \{\mathbf{z} \in \mathbb{R}^n \mid \mathbf{z} \geq 0\}$), the original problem is transformed into:

$$\min_{\mathbf{c}, \mathbf{z}} \frac{1}{2} \|\mathbf{Hc} - \mathbf{b}\|^2 + I_{\mathcal{K}}(\mathbf{z}) \quad \text{s.t.} \quad \mathbf{z} = D(\mathbf{Hc} + \mathbf{f}_0) \tag{5}$$

We incorporate an Exponential Moving Average (EMA) scheme to enhance the ADMM Themelis & Patrinos (2020), which is then used to solve the aforementioned constrained optimization problem.

## 2.3 DUAL-BRANCH CONTRASTIVE LEARNING BASED BTF

We utilize a deep convolutional network to model the BTF, and the BTF proposed in our work consists of a generator network, a discriminator network, and an adaptive feature distribution matching (AFDM) module (shown in Figure 1 (b)).

### 2.3.1 PRE-TRAINING OF CAE

Conventional BTF is strongly associated with CRF; however, as illustrated in Figure 1, BTF and CRF are completely decoupled in our proposed method. Given a non-corresponding pair of images $\mathbf{P}_x$, $\mathbf{P}_y$ in the distorted domain $\mathcal{X} \subset \mathbb{R}^{H \times W \times 3}$ and normal-exposure domain $\mathcal{Y} \subset \mathbb{R}^{H \times W \times 3}$, the irradiance of $\mathbf{P}_x$, $\mathbf{P}_y$ can be denoted as $\mathbf{E}_x$ and $\mathbf{E}_y$. A key advantage of the proposed BTF is that it leverages guide images' irradiance $\mathbf{E}_y$ to generate an implicit representation of exposure ratio. This not only enables flexible control over the enhancement process of $\mathbf{E}_x$ but also enhances robustness when degraded images exhibit domain shift. To construct this high-quality implicit representation, we develop a contrastive learning-based auto-encoder (CAE) and a dual-branch contrastive learning strategy—distinct from conventional approaches to enhance the stability of the learning process and yield highly discriminative features.

The detailed elaboration of our dual-branch contrastive learning algorithm is presented in Figure 2(a). In the $\mathcal{X}$ branch, the query sample $\mathbf{h}$ and positive sample $\mathbf{h}^+$ are both randomly cropped from $\mathbf{E}_x$, while negative samples $\mathbf{h}_i^-$ (where $i \in \{1, 2, \cdots\}$) are image patches randomly cropped from $\mathbf{E}_y$. And the $\mathcal{Y}$ branch adopts the reverse configuration. The query sample $\mathbf{s}$ and positive sample $\mathbf{s}^+$ are both randomly cropped from $\mathbf{E}_y$, while negative samples $\mathbf{s}_i^-$ (where $i \in \{1, 2, \cdots\}$) are image patches randomly cropped from $\mathbf{E}_x$. Utilizing the aforementioned random patches, we can derive the latent feature of them:

$$\{h_j/s_j\}_{j=1}^4 = E_{\mathcal{C}}(\mathbf{h/s}), \ \{\bar{h}_j/\bar{s}_j\}_{j=1}^4 = E_{\mathcal{C}}(\mathbf{h}^+/\mathbf{s}^+), \ \{\tilde{h}_j^i/\tilde{s}_j^i\}_{j=1}^4 = E_{\mathcal{C}}(\mathbf{h}_i^-/\mathbf{s}_i^-) \tag{6}$$

We can derive the contrastive loss term as follows.

$$\mathcal{L}_{\text{info}} = \sum_{j=1}^4 \sum_{i=1}^N \mathbb{E}[\mathcal{L}_{\text{NCE}\Rightarrow\mathcal{X}}(h_j, \bar{h}_j, \tilde{h}_j^i) + \mathcal{L}_{\text{NCE}\Rightarrow\mathcal{Y}}(s_j, \bar{s}_j, \tilde{s}_j^i)] \tag{7}$$

where $\mathcal{L}_{\text{NCE}\Rightarrow\mathcal{X}}(h_j, \bar{h}_j, \tilde{h}_j^i) = -\log \frac{\sigma(h_j, \bar{h}_j)}{\sigma(h_j, \bar{h}_j) + \sum_i \sigma(h_j, \tilde{h}_j^i)}$

The final training loss can be formulated as $\mathcal{L}_{\text{CAE}} = \mathcal{L}_{\text{info}} + \mathcal{L}_{\text{re}}$ where $\mathcal{L}_{\text{re}} = ||\mathbf{E}_x - \widehat{\mathbf{E}}_x||_1 + ||\mathbf{E}_y - \widehat{\mathbf{E}}_y||_1$ where $\widehat{\mathbf{E}}_x$, $\widehat{\mathbf{E}}_y$ represent the reconstructed images of $\mathbf{E}_x$, $\mathbf{E}_y$ through $E_{\mathcal{C}}$, respectively.

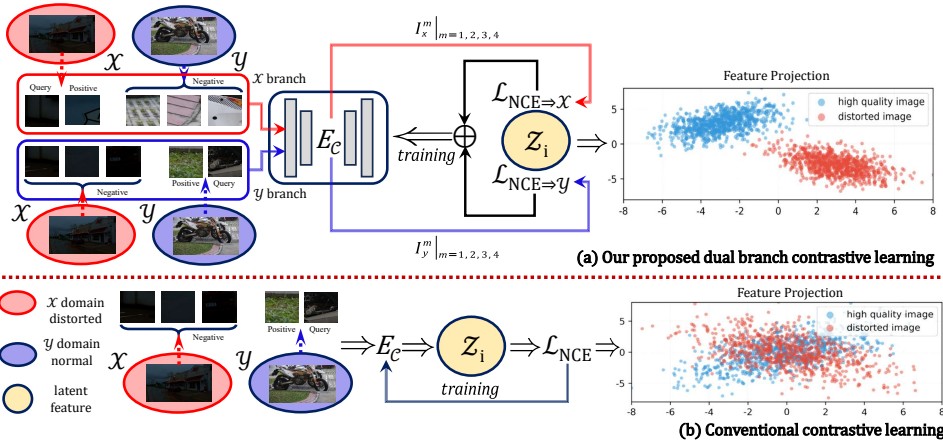

Figure 2: Comparisons of our proposed dual branch contrastive learning and conventional contrastive learning

**Theorem 2.** *The gradient of the contrastive loss term defined in Eq.(7) with respect to the trainable parameters $\theta$ of the contrastive encoder $E_C$ can be decomposed as:*

$$\nabla_\theta \mathcal{L}_{\text{info}} = \frac{1}{\tau} \mathbb{E}\left[\left(1 - p(z^+)\right) \cdot J_\theta^\top(z^+)z - \sum_{i=1}^{K} p(z_i^-) \cdot J_\theta^\top(z_i^-)z\right] \tag{8}$$

*where $z$ denotes the feature representation of the query samples, $z^+$ denotes the feature representation of the positive sample, and $\{z_i\}_{i=1}^{K}$ denote the feature representations of the negative samples. For the sake of simplicity, we denote $z^+$ as $z_{K+1}$. All features are normalized onto the unit hypersphere: $\|z\| = \|z^+\| = \|z_i\| = 1$. $p(z^+) = \frac{\exp\left(z^{+\top}z/\tau\right)}{\sum_{i=1}^{K+1}\exp\left(z_i{}^\top z/\tau\right)}$ stand for the probability that the positive sample is selected, and $p(z_i^-) = \frac{\exp\left(z_i^{-\top}z/\tau\right)}{\sum_{i=1}^{K+1}\exp\left(z_i{}^\top z/\tau\right)}$ is the probability that the i-th negative sample is selected. $J_\theta(z^+) = \frac{\partial z^+}{\partial \theta}$ is the Jacobian matrix of the positive sample features, and $J_\theta(z_i^-) = \frac{\partial z_i^-}{\partial \theta}$ is the Jacobian matrix of the i-th negative sample features.*

**Proof:** The proof can be seen in **Appendix**.

Define the change in the inner product after updating the parameter $\theta$ as $\frac{\partial}{\partial \eta}\langle z^+(\theta + \eta v), z\rangle_{\eta=0}$, where $v = (1/\tau)\mathbb{E}\left[(1 - p(z^+))J_\theta^\top(z^+)z\right]$. Then it can be derived that

$$\begin{aligned}\frac{\partial}{\partial \eta}\langle z^+, z\rangle|_{\eta=0} &= \left\langle \frac{\partial z^+}{\partial \eta}, z\right\rangle = \left\langle J_\theta(z^+)v, z\right\rangle = \left\langle J_\theta(z^+)\mathbb{E}(1 - p(z^+))J_\theta^\top(z^+)z, z\right\rangle \\ &= \mathbb{E}z^\top J_\theta(z^+)(1 - p(z^+))J_\theta^\top(z^+)z = \mathbb{E}(1 - p(z^+))\|J_\theta^\top(z^+)z\|^2 \geq 0\end{aligned} \tag{9}$$

Thus, updating the parameter along this direction will necessarily increase the inner product $\langle z^+, z\rangle$. Moreover, since all features are normalized to the unit hypersphere, an increase in the inner product is equivalent to a decrease in distance, i.e., $z^+$ approaches $z$. Similarly, it can be derived that after updating the parameter $\theta$, the inner product $\langle z_\eta^-, z\rangle$ decreases, and their mutual distance increases.

### 2.3.2 ADAPTIVE FEATURE DISTRIBUTION MATCHING

To characterize the implicit expression of exposure ratio, we propose a AFDM module that can match the eCDFs of multi-level features extracted by CAE. As implied by its name, Sort-Matching algorithm operates through the alignment of two sorted vectors $x$, $y$. ($x : \tau = (\tau_1, \cdots, \tau_n)$, $y : \tau = (\varpi_1, \cdots, \varpi_n)$). The $\{x_{\tau_i}\}_{i=1}^{n}$ and $\{y_{\varpi_i}\}_{i=1}^{n}$ can be regarded as the sorted values of $x$ and $y$ in ascending order. It means that $x_{\tau_1} = \min x$, $x_{\tau_n} = \max x$, $x_{\tau_i} < x_{\tau_j}$ if $i < j$. $\{y_{\varpi_i}\}_{i=1}^{n}$ is

similarly defined. The output $p$ of Sort-Matching can be defined as $p_{\tau_i} = y_{\varpi_i}$ where $p_{\tau_i}$ stand for the $i$-th element of $p$.

We propose AFDM to perform exact eCDFs matching on the basis of Sort-Matching. To enable the gradient back-propagation in deep models, we modify the calculation method of Sort-Matching as follows.

$$I_{xy}^m : o_{\tau_i}^m = \chi_{\tau_i}^m + \rho_m \Upsilon_{\varpi_i}^m - \rho_m \langle \chi_{\tau_i}^m \rangle, \rho_m = E_{\mathcal{A}} \left( I_x^m, I_y^m \right), m = 1, 2, 3, 4 \qquad (10)$$

where $\langle \bullet \rangle$ represents the stop-gradient operation. $I_x^m$, $I_y^m$ denote the $m$-th level feature of image $E_x$ and $E_y$ extracted by the CAE. $[\chi_1^m, \chi_2^m, \cdots, \chi_n^m]$ stand for the 1D vector flatten by feature $\{I_x^m\}_{m=1,2,3,4}$. $[\Upsilon_1^m, \Upsilon_2^m, \cdots, \Upsilon_n^m]$ is the 1D vector flatten by feature $\{I_y^m\}_{m=1,2,3,4}$. The pseudo code of AFDM is shown in **Appendix, Algorithm. 1**.

## 2.4 LOSS FUNCTION

Our proposed method must adhere to physical constraints during the enhanced image generation process. Specifically, the semantic content of the enhanced image must remain consistent with that of the source image. Thus, we design a semantic consistency loss defined as $L_c = \sum_{m=1}^4 ||I_q^m - I_{xy}^m||_2^2$, where $\{I_q^m\}_{m=1,2,3,4} = E_{\mathcal{C}}(f(\mathbf{E}_x))$.

To enable guide images to exert control over the image enhancement process and thereby facilitate domain generalization, we develop a multi-scale InfoNCE loss that maximizes mutual information between the feature distributions of the guide image and the output enhancement across different abstraction levels. Considering a query sample $\mathcal{H}$ that is cropped from enhanced result, its positive sample is $\mathcal{H}^+$ is randomly cropped from $\mathbf{E}_y$, while negative samples $\mathcal{H}^- = \{\mathcal{H}_k^-\}_{k=1,2,\cdots}$ are patches cropped from $\mathbf{E}_x$. The InfoNCE loss can be formulated as follows.

$$L_{mu} = \sum_{i=1}^4 \mathbb{E} \left( \mathcal{L}_{\text{NCE}} \left( I_{\mathcal{H}}^i, I_{\mathcal{H}^+}^i, I_{\mathcal{H}^-}^i \right) \right) \qquad (11)$$

where $I_{\mathcal{H}}^i, I_{\mathcal{H}^+}^i, I_{\mathcal{H}_k^-}^i$ stand for the level $i$ feature of the corresponding images $(\mathcal{H}, \mathcal{H}^+, \mathcal{H}_k^-)$ extracted by CAE $E_{\mathcal{C}}$.

**Theorem 3.** *Minimizing the InfoNCE loss defined in Eq.(11) is equivalent to maximizing the mutual information $I(I_{\mathcal{H}}^i, I_{\mathcal{H}^+}^i)$ between the representations $I_{\mathcal{H}}^i$ and $I_{\mathcal{H}^+}^i$.*

**Proof:** The proof is provided in **Appendix**.

Complementing the generalization of guide features, we introduce a relative adversarial loss $L_{\text{adv}}$ Jolicoeur-Martineau (2018); Mao et al. (2017) that enforce local detail authenticity. The loss function for optimizing BTF is thus written as $\mathcal{L}_{\text{BTF}} = \lambda_c L_c + \lambda_{mu} L_{\text{mu}} + \lambda_{adv} L_{\text{adv}}$.

The pseudocode for the complete training pipeline of the model is provided in the **Appendix**.

# 3 EXPERIMENT

## 3.1 UNDERWATER ROBOTIC VISION: PERCEPTION ENHANCEMENT

To validate the generalization capability of the proposed method in unknown underwater environments, we integrate the proposed CRM into the shore-based processing system of a bridge underwater structural inspection robot. The CRM is employed to enhance the underwater optical images transmitted by the robot, thereby improving its visual perception capabilities.

The underwater robot is equipped with a Sony IMX335 image sensor housed in a watertight enclosure to capture optical images of submerged bridge piers. The acquired images are then transmitted to the shore-based processing center via a fiber-optic umbilical cable. The shore-based processing center is equipped with an NVIDIA RTX 4090 GPU (16GB VRAM) to serve as the computational hardware for executing the proposed algorithm. The BTF training weights are set to $\lambda_c = 2$, $\lambda_{mu} = 1$, $\lambda_{adv} = 2$(Our experimental scenario is shown in Figure 12, Appendix)

Figure 3: Enhancement results of optical images for underwater bridge pier structures I

Table 1: Quantitative comparison of combined low-light and underwater image enhancement methods. Each UIE algorithm is evaluated in two processing orders: LLIE→UIE and UIE→LLIE (pink).

| Metric | HVI-CIDNet | | | | QuadPrior | | | | ZeroIG | | | | Diff-Retinex++ | | | | Ours | RICG | NPreset |
|---|---|---|---|---|---|---|---|---|---|---|---|---|---|---|---|---|---|---|---|
| | WWPF | WFI2-net | TUDA | USUIR | WWPF | WFI2-net | TUDA | USUIR | WWPF | WFI2-net | TUDA | USUIR | WWPF | WFI2-net | TUDA | USUIR | | | |
| UCIQE ↑ | 0.492 / 0.541 | 0.489 / 0.552 | 0.534 / 0.528 | 0.508 / 0.509 | 0.516 / 0.581 | 0.477 / 0.573 | 0.472 / 0.579 | 0.488 / 0.552 | 0.445 / 0.456 | 0.466 / 0.469 | 0.503 / 0.511 | 0.492 / 0.527 | 0.526 / 0.602 | 0.549 / 0.636 | 0.568 / 0.638 | 0.545 / 0.619 | **0.687** | 0.554 | 0.629 |
| UIQM ↑ | 3.492 / 4.129 | 3.117 / 4.083 | 3.825 / 4.338 | 3.712 / 4.262 | 4.114 / 4.562 | 4.206 / 4.583 | 4.188 / 4.669 | 4.003 / 4.413 | 2.973 / 3.266 | 3.026 / 3.260 | 3.084 / 3.311 | 3.141 / 3.308 | 4.107 / 4.886 | 4.226 / 4.757 | 4.338 / 5.026 | 4.255 / 4.926 | **5.462** | 4.671 | 4.928 |
| CCF ↑ | 23.19 / 25.73 | 22.63 / 24.98 | 22.91 / 25.01 | 21.08 / 24.11 | 25.01 / 27.32 | 25.88 / 26.68 | 26.33 / 27.79 | 24.27 / 26.13 | 20.16 / 21.84 | 19.83 / 22.93 | 22.34 / 23.18 | 22.08 / 21.98 | 24.88 / 26.03 | 25.91 / 28.83 | 26.02 / 29.14 | 25.98 / 27.70 | **33.13** | 30.85 | 31.73 |
| US ↑ | 2.17 / 2.29 | 2.23 / 2.26 | 2.05 / 2.19 | 1.82 / 1.89 | 2.44 / 2.53 | 2.59 / 2.65 | 2.58 / 2.63 | 2.20 / 2.50 | 1.22 / 1.31 | 1.09 / 1.29 | 2.08 / 1.45 | 1.91 / 1.33 | 2.47 / 2.53 | 2.51 / 2.61 | 2.61 / 2.68 | 2.60 / 2.62 | **2.83** | 2.26 | 2.55 |

The process begin with the calibration of the CRF for the IMX335 image sensor using our proposed method. Subsequently, unsupervised training of the BTF is conducted using a hybrid dataset of singularly degraded images, formed by combining the SICE dataset Cai et al. (2018) and the LOLv1 dataset Wei et al. (2018). To ensure a fair comparative evaluation, and given that real-world underwater images suffer from concurrent degradations of low light and color cast, our method is compared against the following approaches: a standalone low-light image enhancement algorithm, a standalone underwater image enhancement algorithm, and a combined low-light and underwater enhancement approach. This combined approach encompasses two distinct processing sequences: 1) low-light enhancement followed by color cast correction, and 2) color cast correction followed by low-light enhancement. We evaluate our CRM against those following algorithms: 1) Low-Light Enhancement: **HVI-CIDNet** Yan et al. (2025), **Diff-Retinex++** Yi et al. (2025), **Zero-IG** Shi et al. (2024), **QuadPrior** Wang et al. (2024a), 2) Underwater Enhancement: **WWPF** Zhang et al. (2024), **WFI2-net** Zhao et al. (2024), **TUDA** Wang et al. (2023), **USUIR** Fu et al. (2022). 3) Joint Enhancement: **Neural Preset** Ke et al. (2023), **RICG** Mao & Cui (2024). The visualization results of the comparative experiment for the single algorithms are shown in Figure 3.

As standalone algorithms underperform significantly relative to the CRM, Table 1 summarizes no-reference image quality metrics (UCIQE Yang & Sowmya (2015), UIQM Panetta et al. (2015), CCF Wang et al. (2018) and user study (US)) for enhancement results achieved by combined approaches. The US recruit 29 participants to conduct independent evaluations of enhanced images' visual quality. Each participant received training prior to assessment, focusing on three critical dimensions: 1) presence of over-exposure, under-exposure, or over-enhancement phenomena; 2) introduction of color distortion; and 3) existence of unnatural textures or significant noise artifacts. Visual quality ratings are assessed using a 5-point scale (1-5).

Through visual inspection of the enhanced results in Figure 3 and Table 1, it can be observed that the UIE algorithms demonstrate strong capabilities in correcting color casts but exhibits limited effectiveness in improving overall brightness to restore details in dark regions. In contrast, the LLIE algorithms excel in illumination enhancement but performs poorly in color correction. Only our proposed method can simultaneously address multiple degradation issues while maintaining end-to-end image enhancement, yielding the best visual quality for enhanced images. The enhancement results of the combined methods are shown in **Appendix**, Figure 10 and Figure 11.

## 3.2 EXPERIMENTS ON PUBLIC BENCHMARKS

To demonstrate the proposed method's diverse domain generalization capabilities, this section employs a publicly available dense fog image dataset for testing and validation. The CRM configuration

and parameter settings for testing remain consistent with the subsection 3.1. This experiment utilizes 150 low-light foggy images from RESIDE dataset Li et al. (2018).

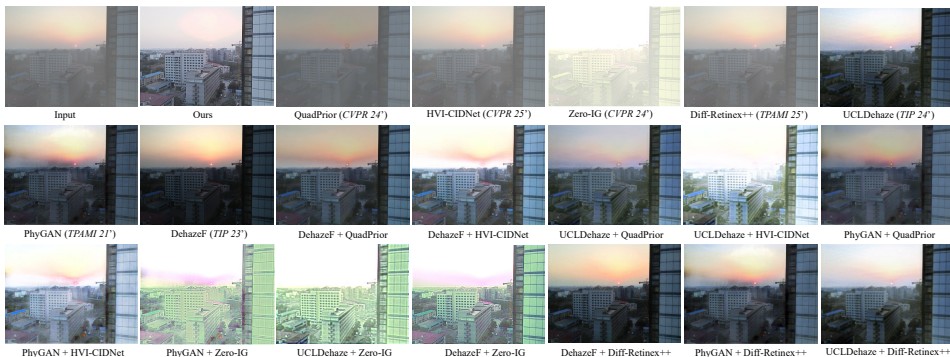

Figure 4: Visualization results on RESIDE dataset Li et al. (2018)

Table 2: Quantitative comparison of combined LLIE and dehazing methods (RESIDE dataset Li et al. (2018)). Each dehazing algorithm is evaluated in two processing orders: LLIE→dehazing and dehazing→LLIE (pink).

| Metric | HVI-CIDNet | | | QuadPrior | | | ZeroIG | | | Diff-Retinex++ | | | Ours |
|---|---|---|---|---|---|---|---|---|---|---|---|---|---|
| | PhyGAN | DehazeF | UCLDehaze | PhyGAN | DehazeF | UCLDehaze | PhyGAN | DehazeF | UCLDehaze | PhyGAN | DehazeF | UCLDehaze | |
| LPIPS ↓ | 0.525 | 0.462 | 0.459 | 0.451 | 0.428 | 0.438 | 0.610 | 0.564 | 0.579 | 0.468 | 0.442 | 0.393 | **0.324** |
| | 0.493 | 0.436 | 0.448 | 0.462 | 0.398 | 0.411 | 0.581 | 0.559 | 0.573 | 0.435 | 0.428 | 0.376 | |
| SSIM ↑ | 0.671 | 0.694 | 0.638 | 0.653 | 0.722 | 0.728 | 0.423 | 0.468 | 0.443 | 0.682 | 0.733 | 0.749 | **0.801** |
| | 0.693 | 0.725 | 0.674 | 0.705 | 0.738 | 0.752 | 0.428 | 0.464 | 0.430 | 0.719 | 0.746 | 0.753 | |
| MUSIQ ↑ | 29.97 | 33.69 | 35.33 | 32.86 | 36.55 | 38.27 | 27.19 | 31.22 | 30.50 | 34.84 | 39.68 | 40.11 | **45.62** |
| | 32.89 | 35.57 | 35.62 | 34.16 | 38.86 | 39.78 | 28.81 | 31.14 | 30.89 | 37.73 | 40.26 | 41.23 | |
| US ↑ | 2.03 | 2.73 | 2.69 | 1.98 | 2.26 | 2.29 | 1.34 | 1.92 | 1.43 | 2.29 | 2.86 | 2.90 | **3.58** |
| | 2.24 | 2.86 | 2.89 | 2.13 | 2.57 | 2.74 | 1.32 | 1.89 | 1.44 | 2.36 | 3.01 | 2.94 | |

We select three state-of-the-art defogging baseline algorithms (**PhyGAN** Pan et al. (2021), **DehazeF** Song et al. (2023), **UCLDehaze** Wang et al. (2024b)) and four state-of-the-art low-light enhancement algorithms (**HVI-CIDNet** Yan et al. (2025), **Diff-Retinex++** Yi et al. (2025), **Zero-IG** Shi et al. (2024), **QuadPrior** Wang et al. (2024a)) for comparison with our method. As shown in the Figure 4, a major drawback of conventional end-to-end methods is their inability to simultaneously address multiple factors such as dense fog, low contrast, and low light conditions that degrade image quality. The majority of combined algorithms, specifically cascaded models, fail to suppress artifact generation and exhibit poor enhancement performance. This is reflected in the quantitative metrics presented in Table 2. Our method outperforms cascaded enhancement models across all evaluation criteria.

## 3.3 CROSS-DEVICE DOMAIN GENERALIZATION

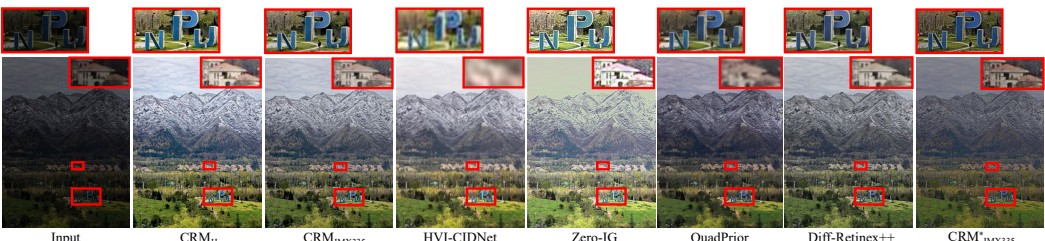

Figure 5: Visualization results on cross-device domain generalization

To evaluate our method's ability to generalize across device domains, we design a dedicated validation experiment. In this experiment, aerial photography is conducted using a DJI Air 3S, with the captured images serving as test samples for the augmentation experiments. We denote the CRM formed by combining the CRF calibrated using the IMX335 with the BTF from the previous sections as $CRM_{IMX335}$. The model combining the CRF calibrated using the DJI Air 3S camera with

the pre-trained BTF is denoted as $\text{CRM}_{\text{H}}$. To validate the impact of CRF calibration error on the model's cross-device domain generalization performance, we employ the method in Grossberg & Nayar (2003) to calibrate the IMX335. The model, formed by combining the pre-trained BTF and CRF calibrated by Grossberg & Nayar (2003), is denoted as $\text{CRM}^*_{\text{IMX335}}$. As shown in the Figure 5, both $\text{CRM}_{\text{H}}$ and $\text{CRM}_{\text{IMX335}}$ effectively restore color and texture details in low-light images. Even when directly utilizing $\text{CRM}_{\text{IMX335}}$, it successfully achieves cross-device domain migration. While CRF calibration errors do not cause the final model to completely lose its intended capabilities, they result in a noticeable decline in its performance regarding cross-device domain generalization. More cross-device domain generalization visualization results of $\text{CRM}_{\text{IMX335}}$ are shown in Figure 17-Figure 18.

### 3.4 ABLATION STUDY

#### 3.4.1 IMPACTS OF DUAL BRANCH CONTRASTIVE LEARNING AND BTF LOSS

To evaluate the impact of the dual-branch contrastive learning strategy of $E_{\mathcal{C}}$ and the training loss term $L_{mu}$ on the model's domain generalization capability, we conduct the following ablation studies: 1) We train the CAE using a conventional contrastive learning strategy; 2) We remove the loss term $L_{mu}$ from the BTF objective function. The resulting models are denoted as $w/o\,\text{dual}$ and $w/o\,L_{mu}$, respectively. The experimental results are shown in Figure 6. The experimental results in Figure 6 (a) and (c) demonstrate that the CRM method incorporating dual-branch contrastive learning achieves lower KL divergence between the enhanced images and the target domain images in RGB kernel density plots, compared to the ablation variant without the dual-branch strategy. This confirms its superior domain generalization performance. The t-SNE visualization in Figure 6(b) demonstrates distinct feature distributions under different mutual information constraints. Guide image features exhibit tight clustering, while enhanced result's features with $L_{mu}$ align closer to the guide distribution, suggesting effective mutual information retention. In contrast, features without this loss form a separate cluster with greater dispersion. The 95% confidence ellipses and centroid connectors quantitatively reinforce this spatial relationship.

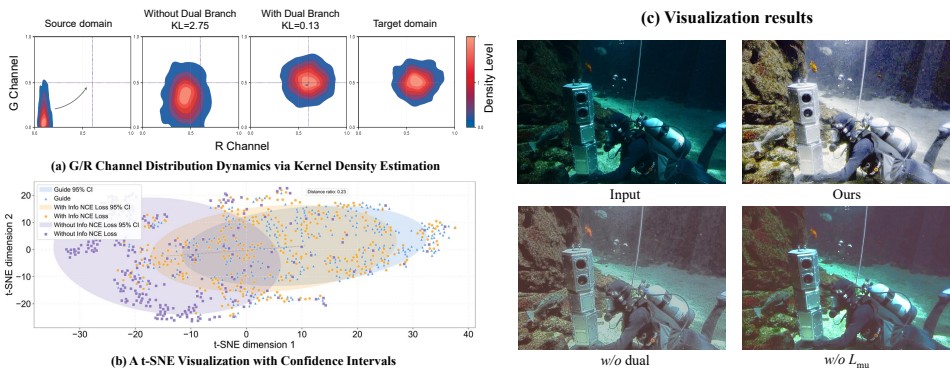

Figure 6: Ablation study on dual-branch contrastive learning and loss $L_{\text{mu}}$

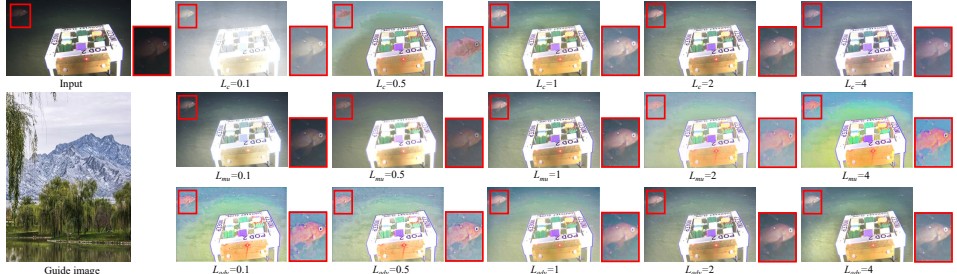

Figure 7: Sensitivity analysis of loss term weight parameters in BTF

We also conduct an ablation analysis on the parameter sensitivities of each loss term in the BTF loss function. In this experiment, we keep the weighting parameters for two loss terms at their final set

values while varying the weighting parameter for the third loss term to 0.1, 0.5, 1, 2, and 4. The specific experimental results are shown in Figure 7. The experimental results indicate that the $L_c$ is most sensitive to the weight parameters. An excessively low value leads to loss of detail in reconstructed images, while an excessively high value impairs the guiding role of the other two loss terms in model training, hindering the restoration of dark region features. The sensitivity of $L_{mu}$ is relatively low in the low-weight parameter range. When the weighting parameter exceeds 1, sensitivity gradually increases. As mentioned earlier, an excessively low lambda value reduces the controllability of the guidance image during the enhancement process, while an excessively high value may cause the tone of the guidance image to adversely affect the enhanced image. The sensitivity to $L_{adv}$ is relatively the lowest, yet its impact remains significant. An excessively low value leads to overly saturated generated images lacking realism, while an excessively high value compromises $L_c$ and generates artifacts. Additionally, we present the SSIM and LPIPS scores for the UIEB dataset under different weight parameter values in **Appendix** Table 2. This further demonstrates that the model trained with $L_c = 2$, $L_{mu} = 1$, $L_{adv} = 2$ achieves the optimal performance.

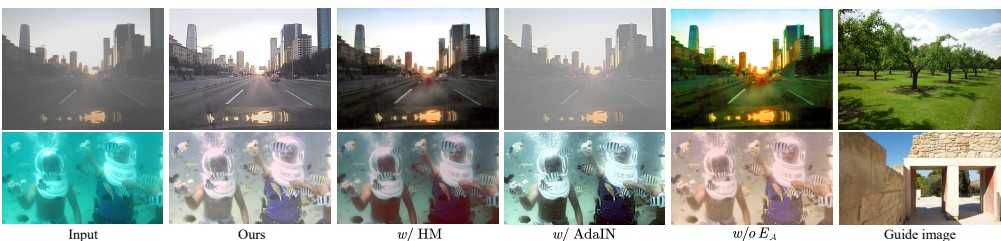

Figure 8: Ablation study on Adaptive Feature Distribution Matching

### 3.4.2 IMPACTS OF ADAPTIVE FEATURE DISTRIBUTION MATCHING

Another essential component of the BTF is the AFDM module, which consists of the Sort-Matching and the encoder $E_{\mathcal{A}}$ that generates adaptive factors. Therefore, it is necessary to conduct ablation studies on both of these elements. In this ablation study, we employ the AdaIN and the histogram matching (HM) method to replace the Sort-Matching for generating mixed features. The resulting models are denoted as $w/\,\mathrm{AdaIN}$ and $w/\,\mathrm{HM}$, respectively. Meanwhile, we remove the encoder $E_{\mathcal{A}}$, causing the AFDM to fully reduce to the traditional Sort-Matching, and denote the resulting trained model as $w/o\,E_{\mathcal{A}}$. As demonstrated in Figure 8, the output weighting coefficients $\rho_m$ generated by $E_{\mathcal{A}}$ endow AFDM with fundamentally distinct characteristics compared to conventional Sort-Matching algorithms. Notably, removing $E_{\mathcal{A}}$ reduces AFDM to traditional Sort-Matching, which undesirably amplifies the tonal influence of guide images on enhancement results.

Real-world data often exhibits substantial deviations from the Gaussian distribution. Consequently, achieving accurate feature distribution alignment through AdaIN becomes challenging. As illustrated in Figure 8, the enhancement results produced by the $w/\,\mathrm{AdaIN}$ fail to effectively handle the multiple degradations induced by domain shift, demonstrating restoration capability only for individual degradation types. As HM assumes a continuous feature distribution, the presence of repeated feature values in those generated by $E_{\mathcal{C}}$ significantly undermines its accuracy in matching the eCDFs. Furthermore, as can be observed from the results in Figure 8, although the $w/\,\mathrm{HM}$ successfully coordinates and addresses the multiple degradations caused by domain shift, its visual quality is also significantly compromised due to the presence of abundant artifacts.

## 4 CONCLUSION

We propose a physics-guided dynamic enhancement framework. Our approach develops a Transformer-based Dynamic CRF with cross-device irradiance calibration, synergized with an AFDM that embeds contrastive learning based multi-scale exposure awareness. This architecture enables implicit exposure ratio estimation by matching eCDFs of arbitrary style references via our CAE, achieving device-agnostic enhancement without manual parameter tuning. Extensive experiments demonstrate our method's superior cross-domain robustness, outperforming state-of-the-art methods on unseen scenarios while preserving natural visual fidelity.

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

| | |
|---|---|
| $\mathbf{P}_x$ | Distorted image |
| $\widehat{\mathbf{P}}_x$ | Enhanced image |
| $\mathbf{P}_y$ | High-visual-quality image |
| $\mathbf{E}_x$ | Image irradiance of $\mathbf{P}_x$ |
| $\mathbf{E}_y$ | Image irradiance of $\mathbf{P}_y$ |
| $\mathcal{X}$ | Distorted domain |
| $\mathcal{Y}$ | Normal-exposure domain |
| $f$ | Camera Response Function (CRF) |
| $g$ | Brightness Transformation Function (BTF) |
| $\mathbf{R}_x$ | Reflectance of $\mathbf{P}_x$ |
| $\mathbf{L}_x$ | Illumination of $\mathbf{P}_x$ |
| $R_i$ | Reflectance of color patch $i$ in Macbeth ColorChecker |
| $S_i$ | Scene radiance of color patch $i$ in Macbeth ColorChecker |
| $E_i$ | Image irradiance of color patch $i$ in Macbeth ColorChecker |
| $\widehat{P}_i$ | Pixel sampling point $i$ on the Macbeth ColorChecker |
| $\widehat{E}_i$ | Relative irradiance sampling point $i$ on the Macbeth ColorChecker |
| $f_0$ | Average curve of the DoRF dataset |
| $q_i$ | The $i$th basis function from the principal component analysis of the curves in the DoRF dataset |
| $\mathbf{h}$ | query sample randomly cropped from $\mathbf{E}_x$ during the training cycle of CAE |
| $\mathbf{h}^+$ | positive sample randomly cropped from $\mathbf{E}_x$ during the training cycle of CAE |
| $\mathbf{h}_i^-$ | The $i$th negative sample randomly cropped from $\mathbf{E}_y$ during the training cycle of CAE |
| $\mathbf{s}$ | query sample randomly cropped from $\mathbf{E}_y$ during the training cycle of CAE |
| $\mathbf{s}^+$ | positive sample randomly cropped from $\mathbf{E}_y$ during the training cycle of CAE |
| $\mathbf{s}_i^-$ | The $i$th negative sample randomly cropped from $\mathbf{E}_x$ during the training cycle of CAE |
| $\mathcal{H}$ | query sample randomly cropped from enhanced result $\widehat{\mathbf{P}}_x$ |
| $\mathcal{H}^+$ | positive sample randomly cropped from $\mathbf{E}_y$ during the training cycle of BTF |
| $\mathcal{H}^-$ | negative sample randomly cropped from $\mathbf{E}_x$ during the training cycle of BTF |
| $E_{\mathcal{C}}$ | Contrastive learning-based auto-encoder (CAE) |
| $E_{\mathcal{A}}$ | Encoder in AFDM |
| $\{I_x^m\}_{m=1,2,3,4}$ | Multi-scale feature of $\mathbf{E}_x$ |
| $\{I_y^m\}_{m=1,2,3,4}$ | Multi-scale feature of $\mathbf{E}_y$ |
| $\{I_{xy}^m\}_{m=1,2,3,4}$ | Multi-scale hybrid features of $\mathbf{E}_x$ and $\mathbf{E}_y$ derived from the AFDM module |
| $\{I_q^m\}_{m=1,2,3,4}$ | Multi-scale feature of $\mathbf{P}_x$ |

## A    TRAINING PSEUDO OF CRM

---

**Algorithm 1:** Training Pseudo of CRM

---

**Input:** Distorted image dataset $\mathbf{X}$, Guide image dataset $\mathbf{Y}$
**Output:** CRM Model parameters $\theta$ (contains parameters of $E_{\mathcal{C}}$, $E_{\mathcal{A}}$ and generator $\mathcal{G}$)

1  Calibrate the CRF $f$ of the robot's onboard image sensor using Eq.(2) to Eq.(5).;
2  $i \leftarrow 1$;
3  **while** $\theta$ *not converged* **do**
4   **if** $i\%3 \neq 0$ **then**
5    Randomly select samples from the datasets $\mathbf{P}_x \leftarrow \mathbf{X}$, $\mathbf{P}_y \leftarrow \mathbf{Y}$;
6    Calculate $\mathcal{L}_{\text{CAE}}$;
7    Update parameters of the $E_{\mathcal{C}}$ using *Adam* optimizer;
8    Freeze the parameters of $E_{\mathcal{C}}$;
9   **else**
10   Randomly select samples from the datasets $\mathbf{P}_x \leftarrow \mathbf{X}$, $\mathbf{P}_y \leftarrow \mathbf{Y}$;
11   image irradiance $\mathbf{E}_x \leftarrow f^{-1}(\mathbf{P}_x)$, $\mathbf{E}_y \leftarrow f^{-1}(\mathbf{P}_y)$;
12   $\{I_x^m\}_{m=1,2,3,4} = E_{\mathcal{C}}(\mathbf{E}_x)$, $\{I_y^m\}_{m=1,2,3,4} = E_{\mathcal{C}}(\mathbf{E}_y)$;
13   $\{I_{xy}^m\}_{m=1,2,3,4} = \text{AFDM}\left(\{I_x^m\}_{m=1,2,3,4}, \{I_y^m\}_{m=1,2,3,4}\right)$;
14   $\widehat{\mathbf{P}}_x \leftarrow f\left(\mathcal{G}\left(\{I_{xy}^m\}_{m=1,2,3,4}\right)\right)$;
15   Calculate $L_{\text{BTF}}\left(\mathbf{P}_x, \widehat{\mathbf{P}}_x, \mathbf{P}_y\right)$;
16   Update parameters of the $\mathcal{G}$, $E_{\mathcal{A}}$ using *Adam* optimizer;
17  **end**
18  $i \leftarrow i + 1$
19 **end**
20 **Return** $\theta$

---

## B    FORWARD REASONING IN THE TESTING PHASE OF THE CRM

---

**Algorithm 2:** Forward Reasoning of CRM

---

**Input:** Arbitary distorted image $\mathbf{P}_x$, Guide image $\mathbf{P}_y$
**Output:** Enhanced image $\widehat{\mathbf{P}}_x$

1  Using the inverse function of the CRF $f$ for a specific camera, input the low-light image $\mathbf{P}_x$ and guide image $\mathbf{P}_y$ to obtain their irradiance $\mathbf{E}_x \leftarrow f^{-1}(\mathbf{P}_x)$, $\mathbf{E}_y \leftarrow f^{-1}(\mathbf{P}_y)$;
2  Obtain multi-scale features of $\mathbf{E}_x$ and $\mathbf{E}_y$ using CAE $E_{\mathcal{C}}$, $\{I_x^m\}_{m=1,2,3,4} = E_{\mathcal{C}}(\mathbf{E}_x)$, $\{I_y^m\}_{m=1,2,3,4} = E_{\mathcal{C}}(\mathbf{E}_y)$;
3  Obtain hybrid features $\{I_{xy}^m\}_{m=1,2,3,4} = \text{AFDM}\left(\{I_x^m\}_{m=1,2,3,4}, \{I_y^m\}_{m=1,2,3,4}\right)$;
4  Obtain the enhanced image $\widehat{\mathbf{P}}_x \leftarrow f\left(\mathcal{G}\left(\{I_{xy}^m\}_{m=1,2,3,4}\right)\right)$;
5  **Return** $\widehat{\mathbf{P}}_x$

---

## C  PSEUDO-CODE FOR AFDM

---

**Algorithm 3:** Pseudo-code for AFDM

---

**Input:** Features $\{I_x^m\}_{m=1,2,3,4}$, $\{I_y^m\}_{m=1,2,3,4}$

**Output:** Mixed feature $\{I_{xy}^m\}^{m=1,2,3,4}$

---

1  $b, c, h, w = I_x^m$.shape;

2  Calculate $\rho_m$ according to Eq.(10);

3  _, IndexforE = torch.sort($I_x^m$.view($b, c$, -1));

4  SortforZ, _ = torch.sort($I_y^m$.view($b, c$, -1));

5  IndexforE = IndexforE.argsort($-1$);

6  $I_{xy}^m = I_x^m + \rho_m$SortforZ.gather(-1,IndexforX) $- \rho_m I_x^m$.detach();

7  **Return** $I_{xy}^m$.view($b, c, h, w$)

---

## D  RELATED WORK

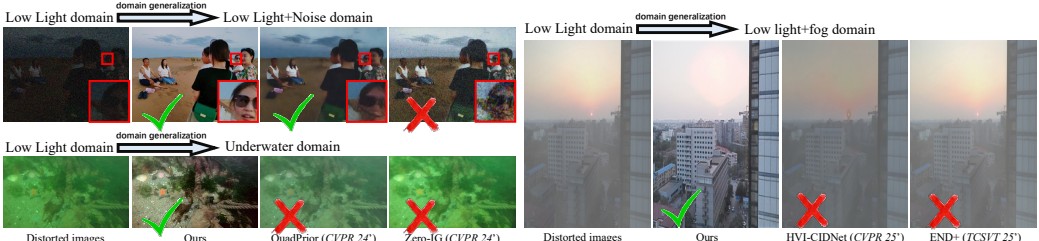

Figure 9: Comparison on domain generalization with a SOTA unsupervised method: Zero-IG Shi et al. (2024), Wang et al. Wang et al. (2024a). All these models are trained on low-light dataset termed LOL Wei et al. (2018). Those three algorithms are generalized in the low-light noise domain, underwater domain, and low-light fog domain. Our model can achieve domain generalization with higher quality than the other two methods.

### D.1  SINGLE DEGRADATION IMAGE ENHANCEMENT

Single degradation image enhancement has advanced through domain-specific physical priors and deep learning integration. In low-light enhancement, Retinex-based decomposition Guo et al. (2016); Xu et al. (2020a) and CNN architectures Wei et al. (2018); Jiang et al. (2021); Liang et al. (2023) address illumination recovery while recent zero-shot methods Guo et al. (2020); Zhang et al. (2021); Ma et al. (2022) optimize without paired data. Underwater restoration combines multi-color space fusion Li et al. (2019; 2021) with real-time designs Islam et al. (2020), whereas denoising leverages residual learning Zhang et al. (2017; 2018) and transformer architectures Liang et al. (2021). Dehazing progresses through atmospheric model reformulation Cai et al. (2016); Li et al. (2017) and multi-scale fusion Dong et al. (2020). However, the practical applicability of these single-degradation image enhancement methods remains limited in real-world engineering scenarios, as natural images often suffer from compounded degradation factors that interact synergistically.

### D.2  CROSS-DOMAIN IMAGE ENHANCEMENT

Recently, many methods have shown impressive performance in cross-domain image enhancement. The Xu et al. (2020b) presents a frequency-aware decomposition-enhancement network that addresses coupled low-light enhancement and noise suppression through layer-specific frequency processing. The Shi et al. (2024) introduces a zero-shot framework for joint low-light enhancement and denoising via illumination-guided physical constraints. The Retinexformer Cai et al. (2023), a one-stage Retinex framework integrating an Illumination-Guided Transformer to jointly address low-light enhancement and noise restoration through non-local interaction modeling. Recent advances in diffusion models Ho et al. (2020); Song et al. (2020) have propelled them to the forefront

of image enhancement research. Wang et al. (2024a); Jiang et al. (2024b; 2023); Yi et al. (2023); Zhou et al. (2023) all fully leverage the progressive denoising capabilities of diffusion models to achieve joint low-light enhancement and denoising. The FourierDiff Lv et al. (2024), a zero-shot framework that integrates Fourier-domain priors with pre-trained diffusion models addresses joint low-light enhancement and deblurring by decoupling amplitude-phase optimization. In Mao et al. (2025); Mao & Cui (2024), the authors develop zero-shot low-light image enhancement methods with underwater color cast correction capability. Although the aforementioned methods possess certain cross-domain image enhancement capabilities, their applicable scenarios remain predetermined and cannot fully detach from cross-domain data during model optimization.

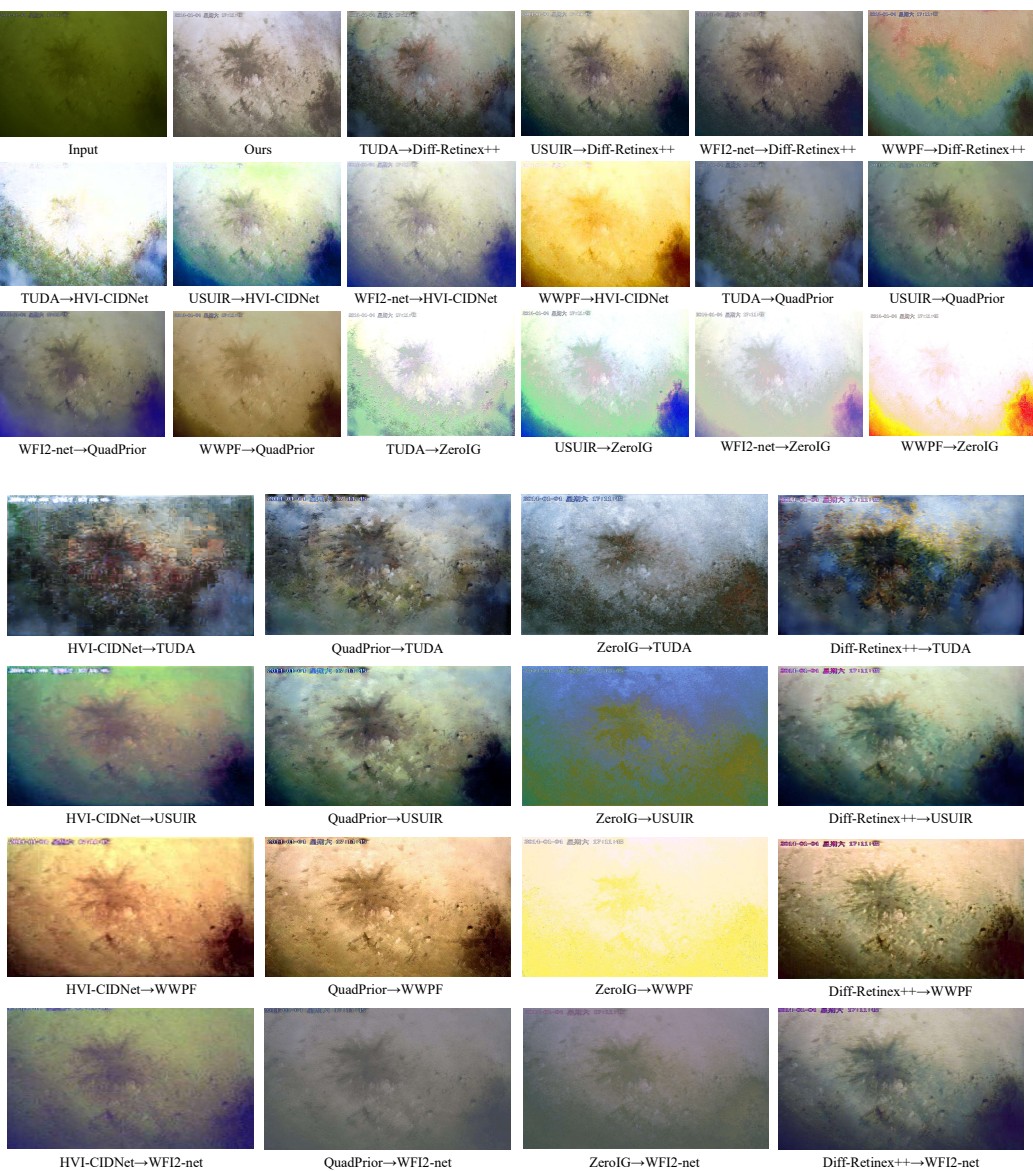

Figure 10: Visualization results of combined approaches I

Input · Ours · TUDA→Diff-Retinex++ · USUIR→Diff-Retinex++ · WFI2-net→Diff-Retinex++ · WWPF→Diff-Retinex++

TUDA→HVI-CIDNet · USUIR→HVI-CIDNet · WFI2-net→HVI-CIDNet · WWPF→HVI-CIDNet · TUDA→QuadPrior · USUIR→QuadPrior

WFI2-net→QuadPrior · WWPF→QuadPrior · TUDA→ZeroIG · USUIR→ZeroIG · WFI2-net→ZeroIG · WWPF→ZeroIG

HVI-CIDNet→TUDA · QuadPrior→TUDA · ZeroIG→TUDA · Diff-Retinex++→TUDA

HVI-CIDNet→USUIR · QuadPrior→USUIR · ZeroIG→USUIR · Diff-Retinex++→USUIR

HVI-CIDNet→WWPF · QuadPrior→WWPF · ZeroIG→WWPF · Diff-Retinex++→WWPF

HVI-CIDNet→WFI2-net · QuadPrior→WFI2-net · ZeroIG→WFI2-net · Diff-Retinex++→WFI2-net

Figure 11: Visualization results of combined approaches II

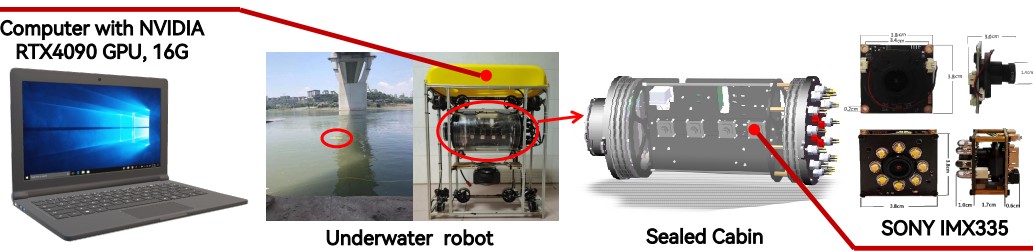

Computer with NVIDIA RTX4090 GPU, 16G

Underwater robot

Sealed Cabin

SONY IMX335

Figure 12: Experiment system of underwater robot

Figure 13: Visualization results on UIEB dataset Li et al. (2019)

# E SUPPLEMENTARY EXPERIMENTAL RESULTS

## E.1 EXPERIMENTAL RESULTS ON UIEB DATASET

This experiment selects 100 low-light underwater images from the UIEB dataset Li et al. (2019) and applies the following two algorithmic approaches for enhancement. The first approach involves single-category enhancement processing, where images are enhanced using either a UIE algorithm or an LLIE algorithm independently. The second approach employs combined enhancement processing, which integrates both UIE and LLIE algorithms to enhance underwater distorted images. We use PSNR, LPIPS, SSIM, MUSIQ, noise level (NL) and US to evaluate the quality of the enhanced results. The scoring criteria for user research and the number of volunteers remain consistent with subsection 3.1. The visual results and quantitative metric evaluations are presented in Figure 13 and Table 3, respectively.

Table 3: Quantitative comparison of combined LLIE and UIE methods (UIEB dataset Li et al. (2019)). Each UIE algorithm is evaluated in two processing orders: LLIE→UIE and UIE→LLIE (pink).

| Metric | HVI-CIDNet | | | | QuadPrior | | | | ZeroIG | | | | Diff-Retinex++ | | | | Ours |
|---|---|---|---|---|---|---|---|---|---|---|---|---|---|---|---|---|---|
| | WWPF | WFI2-net | TUDA | USUIR | WWPF | WFI2-net | TUDA | USUIR | WWPF | WFI2-net | TUDA | USUIR | WWPF | WFI2-net | TUDA | USUIR | |
| LPIPS ↓ | 0.493 | 0.582 | 0.558 | 0.481 | 0.429 | 0.460 | 0.553 | 0.497 | 0.621 | 0.518 | 0.635 | 0.686 | 0.477 | 0.496 | 0.508 | 0.520 | **0.357** |
| | 0.437 | 0.579 | 0.557 | 0.514 | 0.410 | 0.412 | 0.536 | 0.482 | 0.605 | 0.446 | 0.609 | 0.618 | 0.429 | 0.438 | 0.482 | 0.473 | |
| SSIM ↑ | 0.682 | 0.562 | 0.702 | 0.670 | 0.749 | 0.737 | 0.728 | 0.681 | 0.658 | 0.675 | 0.640 | 0.621 | 0.674 | 0.711 | 0.705 | 0.686 | **0.788** |
| | 0.717 | 0.593 | 0.705 | 0.674 | 0.753 | 0.742 | 0.721 | 0.706 | 0.662 | 0.669 | 0.658 | 0.618 | 0.728 | 0.726 | 0.739 | 0.694 | |
| MUSIQ ↑ | 40.26 | 39.77 | 38.75 | 35.68 | 43.36 | 40.02 | 39.87 | 39.96 | 33.83 | 37.14 | 35.61 | 34.78 | 44.79 | 43.16 | 42.83 | 42.10 | **49.86** |
| | 40.99 | 40.08 | 38.82 | 36.62 | 46.11 | 41.17 | 38.16 | 40.27 | 33.98 | 34.26 | 36.39 | 35.64 | 46.12 | 46.74 | 45.25 | 43.96 | |
| NL ↓ | 3.828 | 3.269 | 2.165 | 1.853 | 0.983 | 0.916 | 0.983 | 0.937 | 2.262 | 1.792 | 1.753 | 1.827 | 1.168 | 1.204 | 0.997 | 1.257 | **0.817** |
| | 3.716 | 3.341 | 2.109 | 1.296 | 0.925 | 0.909 | 0.955 | 0.931 | 2.179 | 1.665 | 1.750 | 1.816 | 1.062 | 1.115 | 0.986 | 1.203 | |
| US ↑ | 2.52 | 2.51 | 2.36 | 2.19 | 2.68 | 2.79 | 2.82 | 2.44 | 2.08 | 1.93 | 2.47 | 2.40 | 2.71 | 2.78 | 2.83 | 2.77 | **3.45** |
| | 2.63 | 2.65 | 2.57 | 2.26 | 2.74 | 2.85 | 3.01 | 2.46 | 2.18 | 2.11 | 2.58 | 2.60 | 2.98 | 3.05 | 3.09 | 3.02 | |

Visual experimental results demonstrate that single underwater image enhancement algorithms perform well in correcting underwater color casts, but exhibit limited capability compared to DCRA in addressing feature loss caused by low-light conditions. However, combining LLIE algorithms (e.g., QuadPrior Wang et al. (2024a)) with UIE methods (e.g., HVI-CIDNet Yan et al. (2025), TUDA Wang et al. (2023), and WFI2-net Zhao et al. (2024)) significantly improves the visual quality of enhanced images. Nevertheless, compared to the proposed method, these combined approaches still fall short in terms of authenticity preservation and overall aesthetic improvement. The image quality assessment metrics in the Table 1 also demonstrate that proposed method significantly outperforms all combined methods in terms of human visual perception.

## E.2 NOISY LOW LIGHT DOMAIN GENERALIZATION

To investigate the domain generalization capability of the proposed method under low-light noisy conditions, we continue to utilize Sony IMX335 image sensor to construct the CRF and the BTF is trained by a hybrid dataset formed by combining **SICE** dataset Cai et al. (2018) and **LOLv1**

dataset Wei et al. (2018). Then we conduct quantitative comparative experiments on visual quality and multi-category evaluation metrics using the **UHD-LL** (ultra-high-definition noisy low-light) dataset Li et al. (2023). In this experiment, we include more LLIE algorithms for comparative analysis with the proposed method, primarily including: **QuadPrior** Wang et al. (2024a), **HVI-CIDNet** Yan et al. (2025), **Diff-Retinex++** Yi et al. (2025), **Zero-IG** Shi et al. (2024), **Lighten-Diffusion** Jiang et al. (2024a), **GSAD** Hou et al. (2024), **Retinexformer** Cai et al. (2023), **CLIT-LIP** Liang et al. (2023),**NeRCo** Yang et al. (2023), **UHDFour** Li et al. (2023)..

First, visual comparison results are shown in Figure 14. Experimental results demonstrate that both our proposed CRM and the zero-shot learning framework by **QuadPrior** Wang et al. (2024a) exhibit significant advantages in noise suppression. However, by comparing local texture details in regions (c)(d) of Figure 14, **QuadPrior** Wang et al. (2024a) shows oversaturation phenomenon in brightness correction, verifying the superiority of our approach in preserving image naturalness.

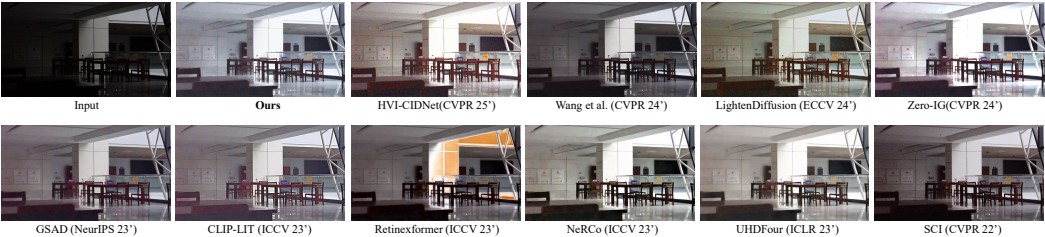

Figure 14: Results on UHD-LL Li et al. (2023)

Table 4: Quantitative Comparative Analysis of State-of-the-Art Methods on the UHD-LL Dataset Li et al. (2023)

| | PSNR↑ | SSIM↑ | LPIPS↓ | MSE(%) | MUSIQ↑ |
|---|---|---|---|---|---|
| UHDFour | 22.19 | 0.860 | 0.208 | 0.303 | 40.55 |
| CLIP-LIT | 20.85 | 0.831 | 0.219 | 0.277 | 38.88 |
| LightenDiffusion | 22.36 | 0.828 | 0.168 | 0.183 | 39.19 |
| QuadPrior | 23.76 | 0.843 | 0.144 | 0.112 | 40.87 |
| HVI-CIDNet | 23.01 | 0.814 | 0.175 | 0.126 | 41.18 |
| Diff-Retinex++ | 22.76 | 0.847 | 0.235 | 0.162 | 37.18 |
| GSAD | 19.76 | 0.742 | 0.314 | 0.264 | 38.92 |
| Retinexformer | 20.71 | 0.773 | 0.323 | 0.225 | 38.26 |
| NeRCo | 23.25 | 0.823 | 0.257 | 0.185 | 41.18 |
| Zero-IG | 18.82 | 0.703 | 0.337 | 0.296 | 32.19 |
| Ours | **24.78** | **0.852** | **0.141** | **0.083** | **45.98** |

For quantitative evaluation, we continue to employ PSNR, SSIM, LPIPS, VIF, NL and US for comprehensive analysis. The user study recruited 29 participants for independent visual quality assessment of enhanced images. The scoring criteria for the US are consistent with those described in Section 3.2 of the main text. The statistical distributions and NL metric correlations are visualized in Figure 15. Quantitative results in Table 4 confirm that CRM achieves superior performance in both noise suppression and perceptual quality when enhancing noisy low-light images, outperforming existing methods across all benchmark metrics.

### E.3 BACKLIT ENVIRONMENT DOMAIN GENERALIZATION

The backlit images tested are selected from the real-world dataset BAID380 Liang et al. (2023), with enhancement results shown in Figure 16. Notably, 4K ultra-high-definition images from this dataset

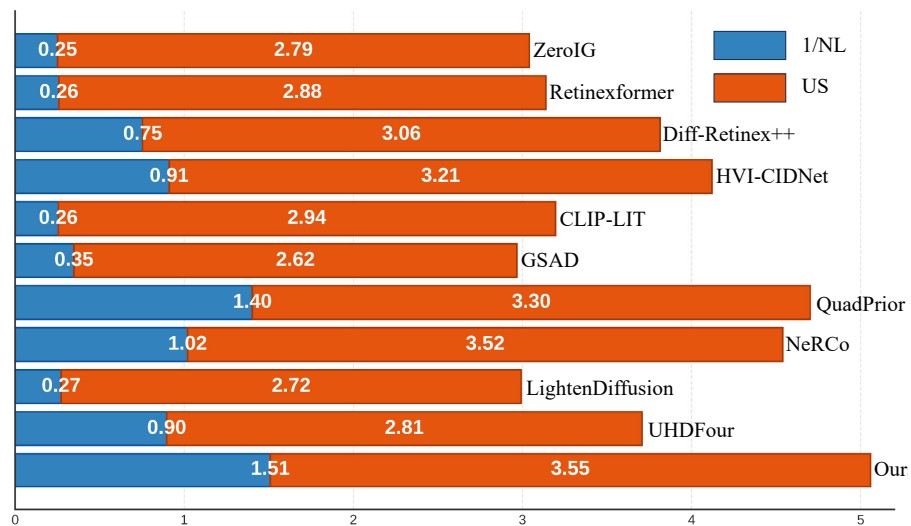

Figure 15: Results on UHD-LL Li et al. (2023)

were chosen as test samples. Experimental results and enlarged regional details demonstrate that the enhanced images generated by our CRM exhibit optimal visual quality. Compared with other algorithms, our method not only effectively improves image brightness but also preserves rich details, while significantly enhancing contrast and mitigating color distortion under backlit conditions.

In comparison, results from **UHDFour** Li et al. (2023) exhibit apparent blurred texture details and overexposure issues, leading to degraded overall image quality. The images generated by **Quad-Prior** Wang et al. (2024a) contain severe artifacts coupled with notable underexposure, further diminishing the visual effectiveness of enhancement. Additionally, **CLIP-LIT** Liang et al. (2023) underperforms in local brightness recovery, failing to adequately improve luminance balance and contrast.

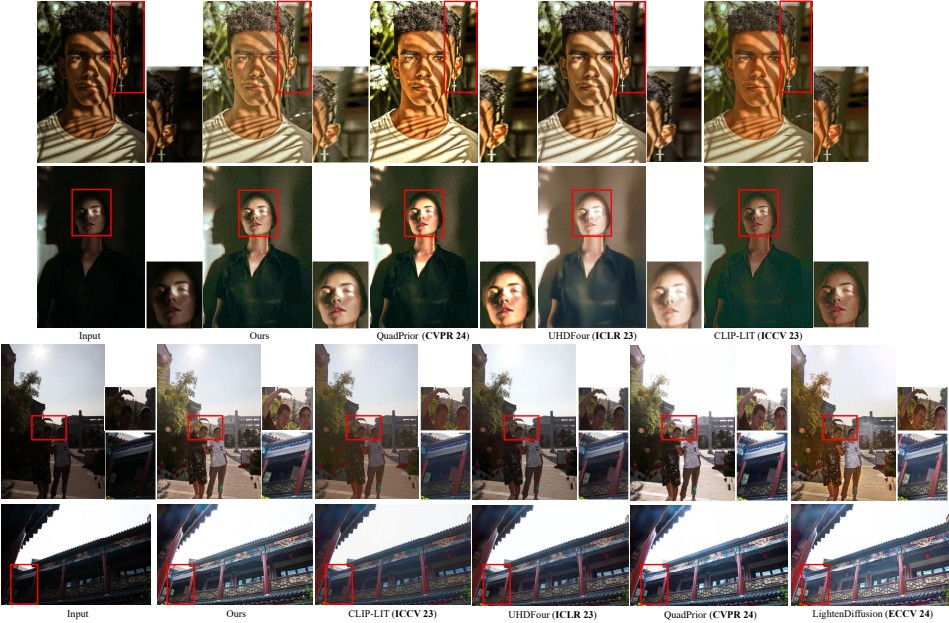

Figure 16: Results on BAID Liang et al. (2023)

The comparative analysis demonstrates that the our CRM effectively reduces overexposure and artifacts while maximally preserving image details under global brightness enhancement. Consequently, our approach demonstrates significant advantages in backlit image enhancement tasks, with generated images outperforming existing state-of-the-art algorithms in visual quality, detail retention, and illumination recovery.

### E.4 CROSS DEVICE GENERALIZATION EXPERIMENT

In wild environments, optical images are often captured by diverse devices. However, existing image enhancement algorithms are typically trained on data from a single device, leading to poor generalization across different device domains. To address this limitation, we collect several low light images using multiple devices (including Huawei Mate 60 Pro, Samsung Galaxy S25 Ultra, Sony $\alpha$7IV, Canon EOS R5, Nikon Z9) to evaluate the cross device domain generalization capabilities of various algorithms. The implementation configurations for the compared LLIE algorithms and our proposed CRM are kept consistent with those described in subsection 3.1, E.3 and E.2. Visual analytical results are obtained through systematic evaluation of images captured by each camera sensor, with experimental results presented in Figure 17 and Figure 18. Experimental data demonstrate that on three flagship camera platforms (Sony $\alpha$7IV, Canon EOS R5, and Nikon Z9), our method exhibits optimal stability, with a cross device standard deviation of 0.08, significantly surpassing other tested methods.

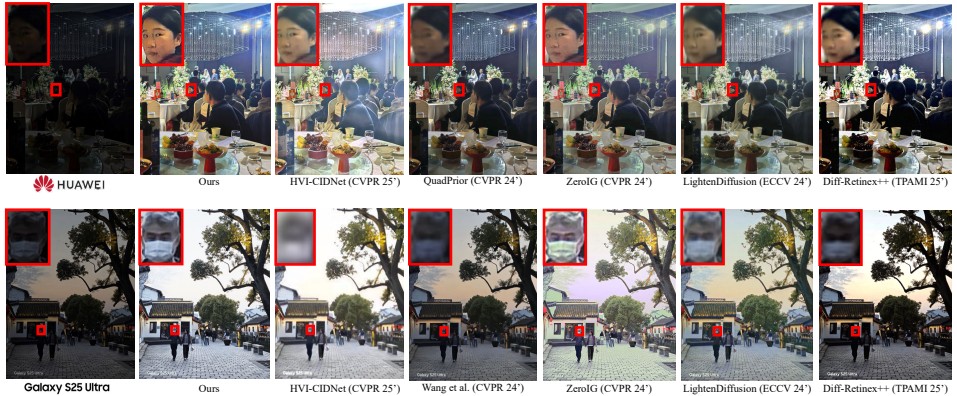

Figure 17: Enhancement Results on Huawei Mate 60pro and Samsung Galaxy S25 Ultra

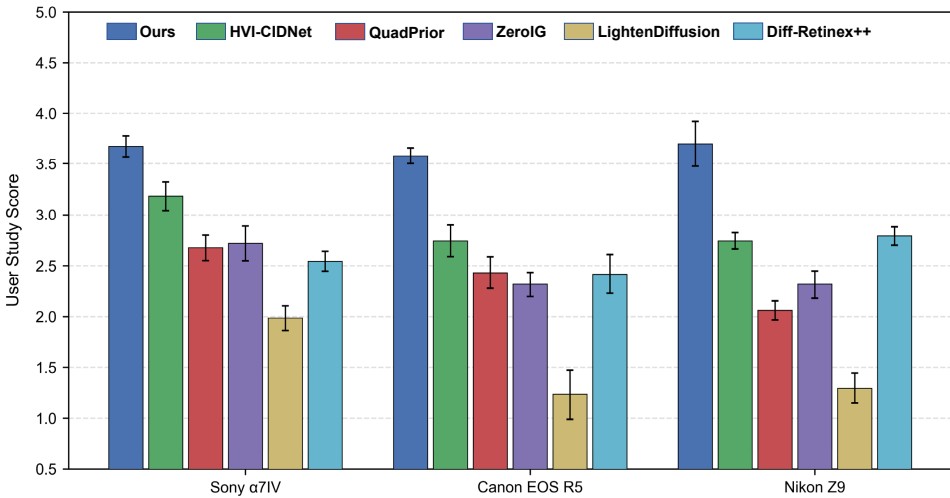

Figure 18: US Scores of Different Algorithms on Images from Different Cameras

### E.5 ABLATION STUDY

#### E.5.1 ABLATION STUDY ON CRF

The ablation study on CRF calibration is divided into two parts. The first part investigate the impact of different CRF curves from DoRF dataset on the enhancement performance of the constructed CRM. The second part examine the importance of the proposed physics based CRF calibration method for obtaining accurate CRF curves. We conduct controlled experiments by selectively applying 6 representative CRFs covering various response characteristics while maintaining identical network architectures and training protocols.

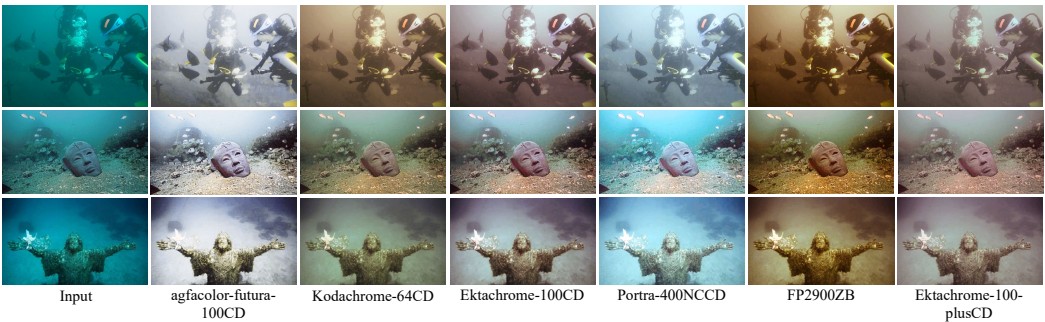

| Input | agfacolor-futura-100CD | Kodachrome-64CD | Ektachrome-100CD | Portra-400NCCD | FP2900ZB | Ektachrome-100-plusCD |

Figure 19: Comparison of enhancement results obtained using different CRFs I

First, the generalization performance experiment in the underwater distortion domain is conducted. The visualized experimental results and the quantitative analysis results are shown in Figure 19 and Table. 5, respectively. Visual comparisons show that the CRM containing agfacolor-futura-100CD's CRF achieves clearer and more accurate correction of color cast distortion, and exhibits superior overall white balance performance. While maintaining natural gradient transitions, this method demonstrates relatively better texture detail preservation compared to the CRFs of other cameras. The quantitative findings corroborate the visual quality comparisons in Figure 19, demonstrating agfacolor-futura-100CD's optimal metric performance while maintaining narrow margins over alternative configurations.

Table 5: Performance Comparison of Six CRF Configurations

| CRF Type | PSNR (dB) | SSIM | CIEDE2000 |
|---|---|---|---|
| agfacolor-futura-100CD | **28.6** | **0.923** | **5.4** |
| Kodachrome-64CD | 28.5 | 0.919 | 5.6 |
| Ektachrome-100CD | 28.3 | 0.917 | 5.7 |
| Portra-400NCCD | 28.1 | 0.914 | 5.9 |
| FP2900ZB | 28.0 | 0.912 | 6.1 |
| Ektachrome-100-plusCD | 27.8 | 0.909 | 6.3 |

The second experiment investigated the generalization capability of models trained with distinct CRF curves from DoRF dataset in low-light noisy domains, utilizing the VV dataset. The comparative results are systematically presented in Figure. 20. Visualization analysis reveals that CRFs exhibit negligible impact on enhancement outcomes for low-illumination noisy domain generalization. While subtle discrepancies in contrast and luminance characteristics persist across configurations, all models achieve consistently satisfactory performance in holistic visual quality restoration.

The third experiment systematically evaluates the cross-domain generalization of models trained with varying CRFs in low-illumination hazy environments. Conducted on the RESIDE benchmark dataset, the comparative performance metrics are quantitatively illustrated in Figure. 21. Experimental results demonstrate consistent efficacy across configurations, where varying CRFs are observed to

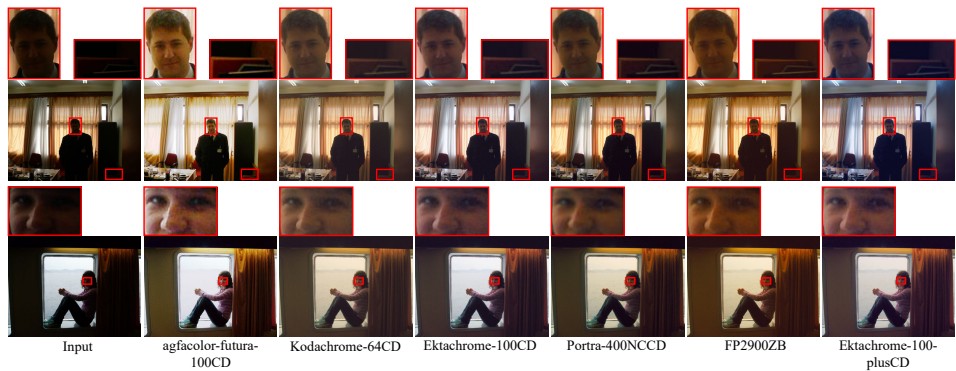

| Input | agfacolor-futura-100CD | Kodachrome-64CD | Ektachrome-100CD | Portra-400NCCD | FP2900ZB | Ektachrome-100-plusCD |

Figure 20: Comparison of enhancement results obtained using different CRFs II

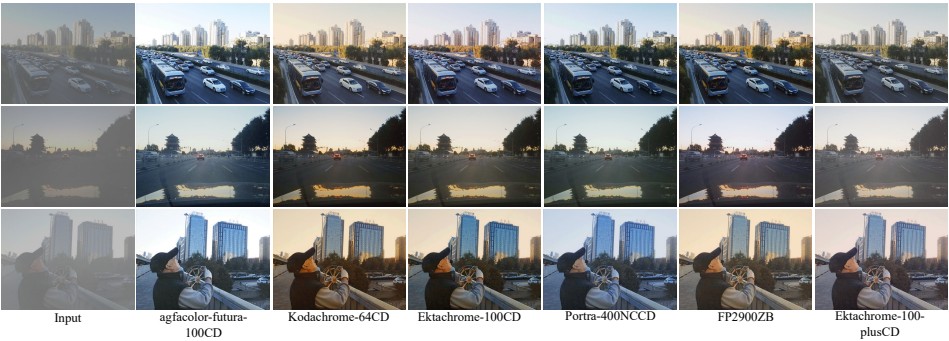

| Input | agfacolor-futura-100CD | Kodachrome-64CD | Ektachrome-100CD | Portra-400NCCD | FP2900ZB | Ektachrome-100-plusCD |

Figure 21: Comparison of enhancement results obtained using different CRFs III

introduce perceptible variations in image stylization while maintaining optimal luminance enhancement and contrast amplification. All models successfully mitigate haze-induced visual degradation, achieving statistically comparable performance metrics as quantified in Figure. 20.

The second ablation study on CRF aims to analyze the impact of both the physics based prior and the EMA-ADMM optimization algorithm in the proposed calibration method on the final results. To provide a clear and intuitive demonstration of the proposed CRF calibration algorithm, we conduct a CRF calibration experiment using a Sony DXC 9000 industrial camera. The calibration results are presented in Figure 22 and Figure 23.

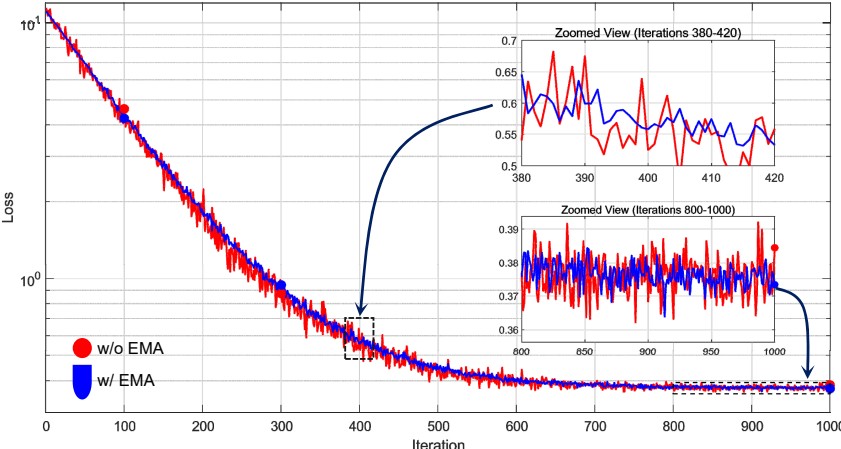

Figure 22: Impact of the exponential moving average mechanism on solving CRF calibration optimization

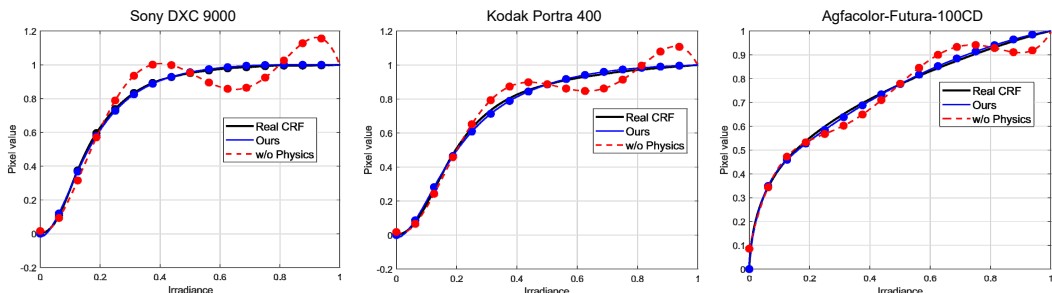

Figure 23: Impact of the physical prior on solving CRF calibration optimization

Figure 22 compares the loss function convergence of EMA-ADMM and traditional ADMM when solving optimization problem Eq.(5). The results clearly demonstrate that the proposed EMA-ADMM algorithm exhibits a significantly more stable optimization process with substantially reduced curve oscillations. Figure 23 presents the ablation study results regarding the proposed physics based prior in CRF calibration. The experiment involves three cameras (a Sony DXC 9000, Kodak Portra 400, and Agfacolor-Futura-100CD) combined with a Macbeth color checker to achieve ground truth CRF calibration. The method from Grossberg & Nayar (2003), which incorporates no physical prior (denoted as $w/o$ Physics), is used for comparison. Experimental results demonstrate that the CRF obtained by our method most closely approximates the ground-truth CRF, while the calibration error of $w/o$ Physics is significantly higher than that of the proposed approach.

### E.5.2 Ablation Study on the choice of the downsampling network in CAE

The proposed CRM is a flexible low-light image enhancement framework. Its flexibility is not only manifested in the controllability of brightness but also in the fact that users can select pre-trained downsampling networks with different structures for CAE $E_C$. To further verify this flexibility, a series of ablation experiments are conducted to test the performance of the model when different downsampling networks are used. Table 6 shows the influence of different downsampling networks on the quality of enhanced images. In this section, several representative downsampling networks are selected for the experiments, among which MobileNet and Inception-ResNet-v2 are the most representative choices.

Table 6: The Influence of Different Downsampling Network Structures on the Efficiency and Performance of the Model(Testing images are selected from BAID dataset Liang et al. (2023))

|  | PSNR↑ | SSIM↑ | LPIPS↓ | FLOPs(G) | Pa(M) | IT(s) |
|---|---|---|---|---|---|---|
| **MobileNet** | 23.45 | 0.822 | 0.183 | **9.46** | **0.33** | **0.0362** |
| SENet | 22.83 | 0.776 | 0.249 | 229.98 | 76.10 | 0.4208 |
| DenseNet | 23.04 | 0.781 | 0.251 | 134.16 | 10.21 | 0.1635 |
| **Inception-ResNet-v2** | **23.88** | **0.898** | **0.143** | 249.63 | 5.19 | 0.2192 |

In application scenarios with high performance requirements, Inception-ResNet-v2 can provide better feature extraction capabilities. Especially in complex scenarios, it can capture more delicate image details and lighting changes. In the task of image enhancement, this network helps to improve the global brightness distribution of the image and retain local details, making it suitable for image enhancement tasks that require high-quality results. However, due to its complex model structure and large computational load, Inception-ResNet-v2 is not very suitable for scenarios with high real-time requirements.

In contrast, MobileNet has become an ideal choice for real-time applications due to its lightweight network structure. In tasks that require efficient processing, MobileNet can significantly reduce the

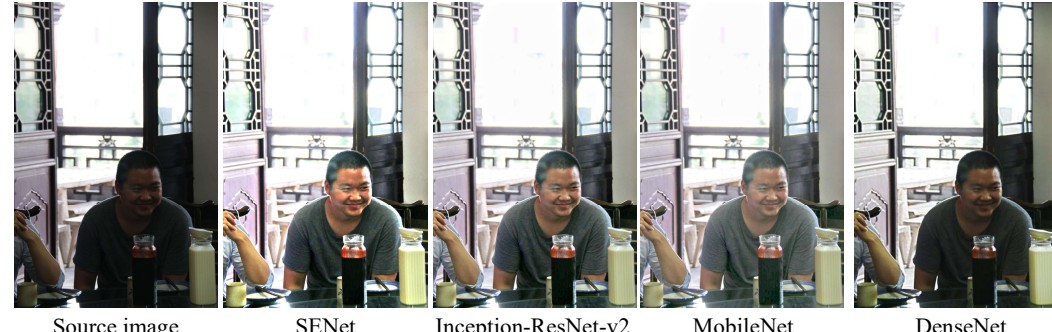

| Source image | SENet | Inception-ResNet-v2 | MobileNet | DenseNet |

Figure 24: Visualized experimental results of the analysis on the influence of downsampling network structures on model performance

consumption of computational resources and provide satisfactory image enhancement effects. Although MobileNet is slightly inferior to Inception-ResNet-v2 in terms of feature expression ability, it can still achieve a good performance trade-off in many practical applications, especially when deployed on embedded systems or devices with limited resources.

Figure 24 shows the visualized comparison results of using MobileNet and Inception-ResNet-v2 as downsampling networks. It can be clearly seen that the enhanced images generated by the models using these two networks both reach a high level of visual quality, and can effectively improve the brightness, contrast, and detail clarity of the images. This indicates that, whether it is used in applications with high-performance requirements or in scenarios with strict real-time requirements, the CRM has sufficient flexibility to meet different needs.

### E.5.3 ANALYSIS OF BRIGHTNESS CONTROL (ABLATION STUDY ON GUIDE IMAGES)

Our proposed method formulates the image enhancement process through two sequential operations: 1) irradiance calibration of low-light images via dynamic CRF mapping, followed by 2) implicit feature extraction from the guidance image's exposure rate through the AFDM module. These extracted features are subsequently decoded to generate enhanced images that maintain exposure consistency with the guide image.

In this ablation experiment, we aim to illustrate the role of our CRM in achieving controllable illumination adjustment through three examples. As shown in Figure 25, distinct guide images present different hues and exposure levels. These are then input into the BTF to produce the final enhanced images. Notably, our CRM effectively transfers the exposure level, instead of the hue, from the guide image to the one to be enhanced, enabling the generation of high-quality enhanced results with any guide image.

As shown in Figure 25, Figure 26 and Figure 27, we select the images from the SICE dataset Cai et al. (2018) and the AGLIE dataset Lv et al. (2021) as input images, and employs the images with different brightness levels from the UHD dataset Li et al. (2023) and the SICE dataset Cai et al. (2018) as guide images. By observing the result graphs, it can be found that the brightness information of the guide images is successfully transferred to the input images, thus achieving precise control over the brightness of the enhanced images. Users can flexibly customize the image brightness according to their requirements. Compared with the existing enhancement methods, our CRM can not only flexibly adjust the brightness but also significantly improve the overall visual quality of the images while maintaining the image details. It can be seen from the results in the figures that the flexible exposure adjustment ability of our CRM effectively overcomes the limitations of the existing methods. Most of the traditional image enhancement methods rely on fixed illumination models and cannot adapt to the complex and changeable lighting conditions in real-world scenarios, especially in extreme environments such as low light and backlight. In contrast, our CRM can dynamically adjust the exposure rate, which not only improves the visual quality of the images but also significantly enhances the detail retention and exposure balance in different scenarios.

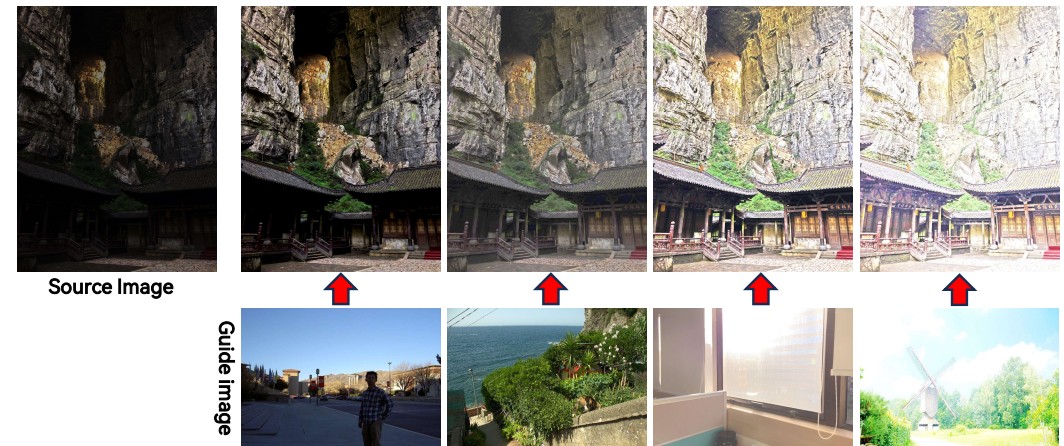

Figure 25: Results of Brightness Control I

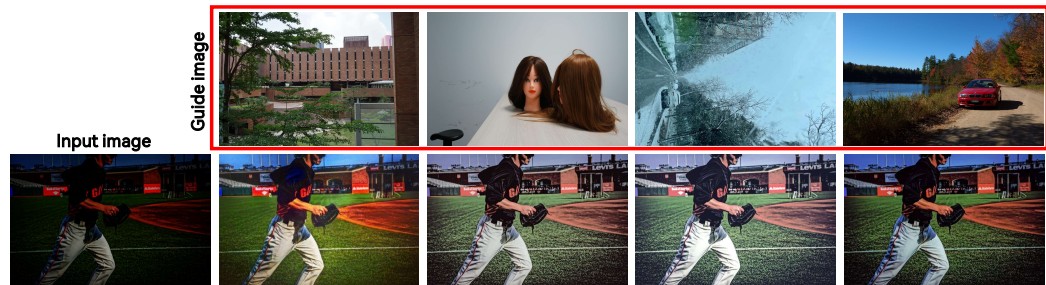

Figure 26: Results of Brightness Control II

In addition, by combining FPN with the BTF, our CRM is not only capable of efficiently addressing the issue of color distortion but also demonstrates strong adaptability in adjusting brightness and contrast. This advantage enables it to achieve domain generalization in image enhancement tasks across a variety of complex environments, such as night scenes, low-light scenarios, and underwater images. In these scenarios, the uneven distribution of light and color deviation often lead to the loss of image details or over-enhancement. However, the guiding mechanism and controllable exposure adjustment strategy of our CRM provide powerful solutions in these aspects, significantly improving the overall visual effect and realism of the enhanced images.

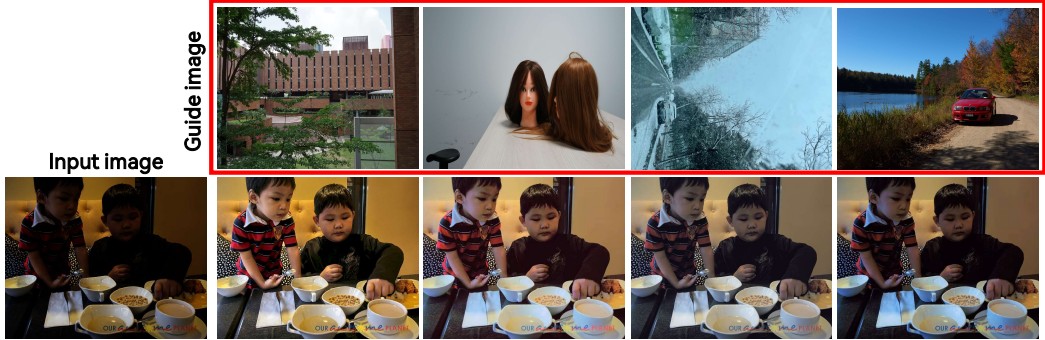

Figure 27: Results of Brightness Control III

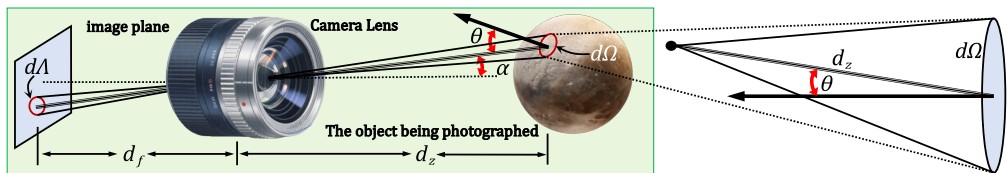

Figure 28: Illustration of solid angle

# F  PROOF

## F.1  PROOF OF THEOREM 1

**Definition 1.** *Solid Angle: Horn (1986) With the observation point as the center, a unit sphere is constructed; the projected area of any object onto this unit sphere is defined as the solid angle of that object with respect to the observation point (The illustration are presented in Figure 28(b)).*

Considering a lens positioned at a distance $d_f$ from the image plane (shown in Figure 28(b)), a small area $\delta\Omega$ on the object surface corresponds to an image area $\delta\Lambda$. The ray from $\delta\Omega$ to the center of the lens forms angles $\alpha$ and $\theta$ with the optical axis and the surface normal, respectively. The object is located at a distance $d_z$ in front of the lens. According to Horn (1986), when light passes through the perspective center without refraction, the solid angle of the cone formed by all rays directed toward a small object area is equal to that of the cone formed by all rays directed toward its corresponding image area. According to the definition of solid angle in **Definition 1**, the solid angles of both the image area and the small object area can be derived as follows.

$$\frac{\delta\Lambda \cos\alpha}{(d_f / \cos\alpha)^2} = \frac{\delta\Lambda \cos^3\alpha}{d_f^2}, \frac{\delta\Omega \cos\theta}{(-d_z / \cos\alpha)^2} = \frac{\delta\Omega \cos\theta \cos^2\alpha}{d_z^2} \tag{12}$$

Since the right-hand sides of the two equations in Eq.(12) are equal, it can be derived as follows.

$$\frac{\delta\Omega}{\delta\Lambda} = \frac{\cos\theta}{\cos\alpha}\left(\frac{d_z}{d_f}\right)^2 \tag{13}$$

Similarly, the solid angle subtended by the lens as viewed from the small area on the photographed object can be formulated as: $\Gamma_{lens} = \frac{\pi r^2 \cos\alpha}{(-d_z / \cos\alpha)^2} = \pi\left(\frac{r}{d_z}\right)^2 \cos^3\alpha$. Let us define the scene radiance as $S$ that is the power emitted per unit projected area per unit solid angle from the photographed object. Therefore, the radiant flux passing through the camera lens from $\delta\Omega$ can be expressed as $\delta Lens = S\Gamma_{lens}\delta\Omega\cos\theta = \pi S\delta\Omega\left(\frac{r}{d_z}\right)^2 \cos^3\alpha\cos\theta$.

Therefore, the irradiance of the image region formed by the light converging from $\delta\Omega$, which is the optical power per unit area, can be expressed as:

$$E = \frac{\delta Lens}{\delta\Lambda} = \pi S\frac{\delta\Omega}{\delta\Lambda}\left(\frac{r}{d_z}\right)^2 \cos^3\alpha\cos\theta = \pi S\left(\frac{r}{d_f}\right)^2 \cos^4\alpha \tag{14}$$

Based on the derivation of the aforementioned physical equations, to accurately calibrate the non-linear relationship between image irradiance and pixel values, we use an LED panel light and a Macbeth ColorChecker as the fundamental experimental setup. During the experiment, the LED panel is positioned in front of the color chart to ensure it illuminated the entire area with uniform brightness. Due to the uniform brightness in the environment, We can derive the relationship between the irradiances $E_i$, $E_j$ of the image regions formed by any two color blocks $i$ and $j$ on the Macbeth ColorChecker, and the scene radiances $S_i$, $S_j$: $\frac{E_i}{E_j} = \frac{\pi\cdot\left(\frac{r}{d_f}\right)^2\cdot\cos^4(\theta_i)\cdot S_i}{\pi\cdot\left(\frac{r}{d_f}\right)^2\cdot\cos^4(\theta_j)\cdot S_j} = \frac{S_i}{S_j}$.

When the spectral power distribution of the light source in the experiment is $I(\lambda)$, according to radiometry Horn (1986); McCluney (2014), the scene radiance $S(\lambda)$ reflected from the surface of

a color patch on a color card can be calculated as: $S(\lambda) = R(\lambda) I(\lambda)$ where $\lambda$ represents the wavelength. For two color patches under the same illumination field, we have

$$\frac{S_i}{S_j} = \frac{R_i \cdot I}{R_j \cdot I} = \frac{R_i}{R_j} \Rightarrow \frac{E_i}{E_j} = \frac{R_i}{R_j} \tag{15}$$

which concludes the proof.

### F.2 PROOF OF THEOREM 2

Prior to proving Theorem 2, we first introduce the following lemma.

**Lemma 4.** *Let the joint distribution $p(\boldsymbol{a}, \boldsymbol{b})$ be symmetric, i.e.,*

$$p(\boldsymbol{a}, \boldsymbol{b}) = p(\boldsymbol{b}, \boldsymbol{a}) \quad \text{for all } \boldsymbol{a}, \boldsymbol{b} \in \mathcal{X}. \tag{16}$$

*where $\mathcal{X}$ denotes a specific probability distribution.*

*Then the following identity holds:*

$$\mathbb{E}_{\boldsymbol{a}, \boldsymbol{b}} \left[ b^\top \frac{\partial a}{\partial \theta} \right] = \mathbb{E}_{\boldsymbol{a}, \boldsymbol{b}} \left[ a^\top \frac{\partial b}{\partial \theta} \right] \tag{17}$$

*where $a = E_{\mathcal{C}}(\boldsymbol{a})$, $b = E_{\mathcal{C}}(\boldsymbol{b})$, $\theta$ represent the trainable parameters in $E_{\mathcal{C}}$.*

*Proof.* We begin by expanding the expectations as integrals:

$$\mathbb{E}_{\boldsymbol{a}, \boldsymbol{b}} \left[ b^\top \frac{\partial a}{\partial \theta} \right] = \iint_{\mathcal{X} \times \mathcal{X}} \left[ E_{\mathcal{C}}(\boldsymbol{b})^\top \frac{\partial E_{\mathcal{C}}(\boldsymbol{a})}{\partial \theta} \right] p(\boldsymbol{a}, \boldsymbol{b}) da db$$
$$\mathbb{E}_{\boldsymbol{a}, \boldsymbol{b}} \left[ a^\top \frac{\partial b}{\partial \theta} \right] = \iint_{\mathcal{X} \times \mathcal{X}} \left[ E_{\mathcal{C}}(\boldsymbol{a})^\top \frac{\partial E_{\mathcal{C}}(\boldsymbol{b})}{\partial \theta} \right] p(\boldsymbol{b}, \boldsymbol{a}) db da \tag{18}$$

Since $p(\boldsymbol{a}, \boldsymbol{b}) = p(\boldsymbol{b}, \boldsymbol{a})$ by the symmetry assumption, we perform a change of variables on the right-hand side of the second equation. Let $u = \boldsymbol{b}$, $v = \boldsymbol{a}$. Then:

$$\mathbb{E}_{\boldsymbol{a}, \boldsymbol{b}} \left[ a^\top \frac{\partial b}{\partial \theta} \right] = \iint_{\mathcal{X} \times \mathcal{X}} \left[ E_{\mathcal{C}}(\boldsymbol{a})^\top \frac{\partial E_{\mathcal{C}}(\boldsymbol{b})}{\partial \theta} \right] p(\boldsymbol{a}, \boldsymbol{b}) da db$$
$$= \iint_{\mathcal{X} \times \mathcal{X}} \left[ E_c(v)^\top \frac{\partial E_c(u)}{\partial \theta} \right] p(v, u) dv du \tag{19}$$

Using the symmetry $p(v, u) = p(u, v)$, we obtain:

$$\mathbb{E}_{\boldsymbol{a}, \boldsymbol{b}} \left[ a^\top \frac{\partial b}{\partial \theta} \right] = \iint_{\mathcal{X} \times \mathcal{X}} \left[ E_c(v)^\top \frac{\partial E_c(u)}{\partial \theta} \right] p(u, v) du dv \tag{20}$$

Since $u$ and $v$ are dummy variables, we may rename them back to $\bar{a}$ and $a$, respectively:

$$\iint_{\mathcal{X} \times \mathcal{X}} \left[ E_{\mathcal{C}}(v)^\top \frac{\partial E_c(u)}{\partial \theta} \right] p(u, v) du dv = \iint_{\mathcal{X} \times \mathcal{X}} \left[ E_{\mathcal{C}}(\boldsymbol{b})^\top \frac{\partial E_{\mathcal{C}}(\boldsymbol{a})}{\partial \theta} \right] p(\boldsymbol{b}, \boldsymbol{a}) da db \tag{21}$$

Therefore, we conclude:

$$\mathbb{E}_{\boldsymbol{a}, \boldsymbol{b}} \left[ a^\top \frac{\partial b}{\partial \theta} \right] = \iint_{\mathcal{X} \times \mathcal{X}} \left[ E_{\mathcal{C}}(\boldsymbol{b})^\top \frac{\partial E_{\mathcal{C}}(\boldsymbol{a})}{\partial \theta} \right] p(\boldsymbol{b}, \boldsymbol{a}) da db = \mathbb{E}_{\boldsymbol{a}, \boldsymbol{b}} \left[ b^\top \frac{\partial a}{\partial \theta} \right] \tag{22}$$

which completes the proof. $\square$

Based on **Lemma 4**, we now proceed to the proof of Theorem 2 To simplify the notation, we define the variables $s^+$ and $s_i^-$ as follows.

$$s^+ = \frac{1}{\tau} z^\mathrm{T} z^+, \quad s_i^- = \frac{1}{\tau} z^\mathrm{T} z_i^- \quad \text{for } i = 1, \ldots, K \tag{23}$$

The loss function can then be rewritten as:

$$\mathcal{L}_\theta = \mathbb{E}\left[-\log\frac{e^{s^+}}{e^{s^+} + \sum_{i=1}^K e^{s_i^-}}\right] = \mathbb{E}\left[-s^+ + \log\left(e^{s^+} + \sum_{i=1}^K e^{s_i^-}\right)\right] \tag{24}$$

Define the scalar function:

$$\mathscr{L}_\theta(s^+, s_1^-, \ldots, s_K^-) = -s^+ + \log\left(e^{s^+} + \sum_{i=1}^K e^{s_i^-}\right) \tag{25}$$

By the multivariate chain rule, the gradient of the loss with respect to the parameters $\theta$ is:

$$\nabla_\theta \mathcal{L} = \mathbb{E}\left[\frac{\partial \mathcal{L}}{\partial s^+} \cdot \frac{\partial s^+}{\partial \theta}\right] + \mathbb{E}\left[\sum_{i=1}^K \frac{\partial \mathcal{L}}{\partial s_i^-} \cdot \frac{\partial s_i^-}{\partial \theta}\right] \tag{26}$$

From Eq.(24), the partial derivatives with respect to $s^+$ and $s_i^-$ are computed as:

$$\frac{\partial \mathscr{L}}{\partial s^+} = \left[-1 + \frac{e^{s^+}}{e^{s^+} + \sum_{j=1}^K e^{s_i^-}}\right] = p(z^+) - 1, \quad \frac{\partial \mathscr{L}}{\partial s_i^-} = \frac{e^{s_i^-}}{e^{s^+} + \sum_{j=1}^K e^{s_j^-}} = p(z_i^-) \tag{27}$$

where $p(z^+)$ and $p(z_i^-)$ are defined as in the theorem.

Finally, the gradients of $s^+$ and $s_i^-$ with respect to $\theta$ are given by:

$$\begin{aligned}
\frac{\partial s^+}{\partial \theta} &= \frac{1}{\tau}\left(z^\top \frac{\partial z^+}{\partial \theta} + (z^+)^\top \frac{\partial z}{\partial \theta}\right) \\
\frac{\partial s_i^-}{\partial \theta} &= \frac{1}{\tau}\left(z^\top \frac{\partial z_i^-}{\partial \theta} + (z_i^-)^\top \frac{\partial z}{\partial \theta}\right)
\end{aligned} \tag{28}$$

Substituting Eq.(28) into Eq.(26), we obtain:

$$\nabla_\theta \mathcal{L} = \frac{1}{\tau}\mathbb{E}\left[(p(z^+) - 1)\left(z^\top \frac{\partial z^+}{\partial \theta} + (z^+)^\top \frac{\partial z}{\partial \theta}\right)\right] + \frac{1}{\tau}\sum_{i=1}^K \mathbb{E}\left[p(z_i^-)\left(z^\top \frac{\partial z_i^-}{\partial \theta} + (z_i^-)^\top \frac{\partial z}{\partial \theta}\right)\right] \tag{29}$$

Expanding the terms yields:

$$\nabla_\theta \mathcal{L} = \frac{1}{\tau}\mathbb{E}\left[(p(z^+) - 1)z^\top \frac{\partial z^+}{\partial \theta}\right] + \frac{1}{\tau}\mathbb{E}\left[(p(z^+) - 1)(z^+)^\top \frac{\partial z}{\partial \theta}\right] \tag{30}$$

$$+ \frac{1}{\tau}\sum_{i=1}^K \mathbb{E}\left[p(z_i^-)z^\top \frac{\partial z_i^-}{\partial \theta}\right] + \frac{1}{\tau}\sum_{i=1}^K \mathbb{E}\left[p(z_i^-)(z_i^-)^\top \frac{\partial z}{\partial \theta}\right] \tag{31}$$

Since the query sample $z$ and the positive sample $z^+$ are independently drawn from the same distribution, their joint distribution is symmetric, i.e., $p(z, z^+) = p(z^+, z)$. According to **Lemma 4**, we have $\mathbb{E}_{z,z^+}\left[(z^+)^\top \frac{\partial z}{\partial \theta}\right] = \mathbb{E}_{z,z^+}\left[z^\top \frac{\partial z^+}{\partial \theta}\right]$.

In contrast, traditional cross-domain negative samples do not share the same marginal distribution as the query sample, so their joint distribution does not satisfy symmetry. Under the proposed dual-branch contrastive learning framework:

(1) In the first branch, the query sample satisfies $z \sim \mathcal{X}$, the positive sample $z^+ \sim \mathcal{X}$, and the negative samples $z_i^- \sim \mathcal{Y}$.

(2) In the second branch, the query sample satisfies $z \sim \mathcal{Y}$, the positive sample $z^+ \sim \mathcal{Y}$, and the negative samples $z_i^- \sim \mathcal{X}$.

This construction enforces symmetry in the joint distribution between query and negative samples across branches, hence:

$$\mathbb{E}_{z,z_i^-}\left[(z_i^-)^\top \frac{\partial z}{\partial \theta}\right] = \mathbb{E}_{z,z_i^-}\left[z^\top \frac{\partial z_i^-}{\partial \theta}\right] \tag{32}$$

Using the identity $p(z^+) - 1 = -(1 - p(z^+))$, the update direction $-\nabla_\theta \mathcal{L}$ simplifies to:

$$-\nabla_\theta \mathcal{L} = \frac{1}{\tau}\mathbb{E}\left[(1 - p(z^+)) \cdot \left(\frac{\partial z^+}{\partial \theta}\right)^\top z - \sum_{i=1}^{K} p(z_i^-) \cdot \left(\frac{\partial z_i^-}{\partial \theta}\right)^\top z\right] \tag{33}$$

Let $J_\theta(z) = \frac{\partial z}{\partial \theta}$ denote the Jacobian matrix. Then we can write:

$$-\nabla_\theta \mathcal{L} = \frac{1}{\tau}\mathbb{E}\left[(1 - p(z^+)) \cdot J_\theta(z^+)^\top z - \sum_{i=1}^{K} p(z_i^-) \cdot J_\theta(z_i^-)^\top z\right] \tag{34}$$

This completes the proof.

**Corollary 1.** *The gradient decomposition formula has a clear geometric interpretation:*

*1. Positive Sample Attraction Term:* $(1 - p(z^+)) \cdot J_\theta^\top(z^+)z$. *The weight factor $1 - p(z^+)$ increases as the similarity between the anchor and the positive sample decreases. Updating the encoder parameter $\theta$ drives the multi scale features of $z^+$ and $z$ to approach, which serves to reduce intra class distance and enhance feature invariance by design.*

*2. Negative Sample Repulsion Term:* $-\sum_{i=1}^{K} p(z_i^-) \cdot J_\theta^\top(z_i^-)z$. *The weight factor $p(z_i^-)$ grows with higher similarity between the anchor and negative samples. Optimizing the encoder parameter $\theta$ enforces the multi scale features of $z_i^-$ and $z$ to diverge, achieving the effect of increasing inter - class distance and improving feature discriminability.*

F.3    PROOF OF THEOREM 3

To simplify the notation in the proof, we denote $I_{\mathcal{H}^+}^i$ from enhanced image and $\{I_{\mathcal{H}_k^-}^i\}_{k=1}^M$ from irradiance of low light image as a set $\mathcal{X} = \{X_1, X_2, \ldots, X_{M+1}\}$ of $M+1$ samples, which includes one positive sample and $M$ negative samples. Following the analysis in Oord et al. (2018), the expression

$$\mathbb{E}\left[-\log \frac{\sigma(I_{\mathcal{H}^+}^i, I_{\mathcal{H}}^i)}{\sigma(I_{\mathcal{H}^+}^i, I_{\mathcal{H}}^i) + \sum_{k=1}^{M} \sigma(I_{\mathcal{H}_k^-}^i, I_{\mathcal{H}}^i)}\right]$$

represents the cross-entropy loss for correctly classifying the positive sample within the set $X$ where $I_{\mathcal{H}}^i$ denotes the query sample from guide image.

Define the event $[G_t = 1]$ to indicate that the sample $X_t$ drawn from $X$ is the positive sample, and $[G_t = 0]$ to indicate that $X_t$ is a negative sample. Then, the optimal probability for this cross-entropy loss is given by the posterior probability $p(G_t = 1 \mid X_t, I_{\mathcal{H}}^i)$. Minimizing this cross-entropy classification loss

$$\min -\mathbb{E}\log\left[p\left(G_t = 1 | X_t, I_{\mathcal{H}}^i\right)\right]$$

is equivalent to maximizing the likelihood of $p(G_t = 1 \mid X_t, I_{\mathcal{H}}^i)$.

Since the set contains $M$ negative samples and one positive sample, the prior probabilities are given by:

$$P(G_t = 1) = \frac{1}{M+1}, \quad P(G_t = 0) = \frac{M}{M+1}.$$

Using Bayes' theorem, the posterior probability $P(G_t = 1 \mid X_t, I_{\mathcal{H}}^i)$ can be expressed as:

$$
\begin{aligned}
p\left(G_t = 1 | X_t, I_{\mathcal{H}}^i\right) &= \frac{p\left(X_t, I_{\mathcal{H}}^i | G_t = 1\right) p\left(G_t = 1\right)}{p\left(X_t, I_{\mathcal{H}}^i | G_t = 1\right) p\left(G_t = 1\right) + p\left(X_t, I_{\mathcal{H}}^i | G_t = 0\right) p\left(G_t = 0\right)} \\
&= \frac{p\left(X_t, I_{\mathcal{H}}^i | G_t = 1\right) \frac{1}{M+1}}{p\left(X_t, I_{\mathcal{H}}^i | G_t = 1\right) \frac{1}{M+1} + p\left(X_t, I_{\mathcal{H}}^i | G_t = 0\right) \frac{M}{M+1}} \\
&= \frac{p\left(X_t, I_{\mathcal{H}}^i | G_t = 1\right)}{p\left(X_t, I_{\mathcal{H}}^i | G_t = 1\right) + (M) p\left(X_t, I_{\mathcal{H}}^i | G_t = 0\right)}
\end{aligned}
\tag{35}
$$

Since the positive sample and $I_{\mathcal{H}}^i$ are conditionally independent given the event, we have $p(X_t, I_{\mathcal{H}}^i \mid G_t = 1) = p(X_t \mid I_{\mathcal{H}}^i)$. Meanwhile, under the assumption that the negative samples are independent of $I_{\mathcal{H}}^i$, it follows that:

$$
p(X_t, I_{\mathcal{H}}^i \mid G_t = 0) = p(X_t) \cdot p(I_{\mathcal{H}}^i).
$$

Thus Eq.(35) can be further simplified into:

$$
p\left(G_t = 1 | X_t, I_{\mathcal{H}}^i\right) = \frac{p\left(X_t, I_{\mathcal{H}}^i\right)}{p\left(X_t, I_{\mathcal{H}}^i\right) + (N-1) p\left(X_t\right) p\left(I_{\mathcal{H}}^i\right)}
\tag{36}
$$

Based on the conclusion derived from Eq.(36), the loss function $L_{mu}$ can be rewritten as follows.

$$
\begin{aligned}
\mathcal{L}_{\mathrm{mu}} &= \sum_{i=1}^{4} -\mathbb{E} \log\left[\frac{p(X_t, I_{\mathcal{H}}^i)}{p(X_t, \phi_i(k)) + (N-1)p(X_t)p(I_{\mathcal{H}}^i)}\right] \\
&= \sum_{i=1}^{4} \mathbb{E} \log\left[\frac{p(X_t, I_{\mathcal{H}}^i) + Mp(X_t)p(I_{\mathcal{H}}^i)}{p(X_t, I_{\mathcal{H}}^i)}\right] \\
&= \sum_{i=1}^{4} \mathbb{E} \log\left[1 + M\frac{p(X_t)p(I_{\mathcal{H}}^i)}{p(X_t, I_{\mathcal{H}}^i)}\right] \\
&= \sum_{i=1}^{4} \mathbb{E} \log\left[1 + M\frac{p(X_t)}{p(X_t | I_{\mathcal{H}}^i)}\right] \\
&> \sum_{i=1}^{4} \mathbb{E} \log\left[M\frac{p(X_t)}{p(X_t | I_{\mathcal{H}}^i)}\right] \\
&= 4 \log[M] + \sum_{i=1}^{4} \mathbb{E} \log\left[\frac{p(X_t)}{p(X_t | I_{\mathcal{H}}^i)}\right]
\end{aligned}
\tag{37}
$$

By the definition of mutual information, we have:

$$
\sum_{i=1}^{4} I\left(X_t, I_{\mathcal{H}}^i\right) > 4 \log[M] - L_{mu}
\tag{38}
$$

Therefore, minimizing the loss function $L_{mu}$ is equivalent to increasing a lower bound on the mutual information $\sum_{i=1}^{4} I\left(X_t, I_{\mathcal{H}}^i\right)$ between the reflection component and the positive sample. This completes the proof.

