# OpenReview forum: "Mastering Domain Shift Image Enhancement Via Differentiable Physics"
_ICLR.cc/2026/Conference — Submitted to ICLR 2026_

### Official Review · Reviewer_vfWa · 2025-10-30

**Soundness:** 3
**Presentation:** 3
**Contribution:** 3
**Rating:** 6
**Confidence:** 2

**Summary:**

This paper tackles image enhancement generalization across domains that exhibit diverse and coupled degradations. The authors decouple CRF and BTF, and leverage contrastive learning together with a generative network. First, CRF calibration is performed by modeling camera nonlinearity with DoRF-based bases. Next, representations are learned in irradiance space via contrastive learning between a distorted domain and a normal-exposure domain. Finally, an AFDM module is trained using a value-magnitude–based sort-matching algorithm, enabling robust enhancement on unseen domains.

**Strengths:**

1. Presents a framework capable of handling complex and unseen degradations that prior LLIE/UIE approaches typically struggle with.
2. Clearly articulates that the conventional CRM coupling between CRF and BTF can hinder enhancement under domain shift, and proposes a principled decoupling.
3. Utilizes dual-branch contrastive learning to capture domain representations in irradiance space.
4. The paper offers dense analyses and thorough empirical comparisons, supporting the validity of the proposed methodology and aiding readability.

**Weaknesses:**

1. The paper would benefit from an explicit assessment of CRF optimization quality and a sensitivity analysis quantifying the impact of CRF estimation errors on enhancement outcomes.
2. Implementation details are limited (e.g., network specifications, hyperparameters, and experimental settings), which may impede reproducibility.

**Questions:**

1. In the absence of any camera configuration information during training, is CRF calibration still feasible?
2. When constructing distorted vs. normal-exposure domains, must the datasets be captured with the same camera and/or contain similar content (e.g., underwater imagery vs. everyday scenes), or is cross-camera/cross-content pairing acceptable?
3. In AFDM, what is the precise rationale for value-based sorting? Given that deeper layers encode different semantics, why is a uniform sorting mechanism appropriate across layers?
4. Do you anticipate the framework to remain effective when the distorted/normal domain pair extends beyond low-light scenarios?
5. Were hyperparameter studies conducted for the loss weights composing L_BTF? If so, please summarize the settings and findings.

---

> ### Author Response · Authors · 2025-11-20
> **Reply to the Question**
>
> Response to Questions 1:
>
> Yes, CRF calibration remains feasible without prior camera configuration. Our method relies on physical radiometric principles rather than specific camera settings.
>
> We derive the mathematical relationship between reflectance in the Macbeth ColorChecker tiles and scene irradiance based on radiometric principles. Consequently, without camera configuration data, we can achieve CRF calibration for any new camera model using the EMoR framework. This is accomplished by capturing images of the Macbeth ColorChecker tiles under uniform illumination, collecting multiple irradiance and pixel value samples, and applying the derived mathematical relationship.
>
> We present the corresponding results in the Appendix E.5.1, Figure 23. Using cameras such as the Sony DXC9000 available in our laboratory, we conducted CRF calibration experiments and achieved high-precision curve fitting without any camera configuration information.
>
> Response to Questions 2:
>
> When constructing the distortion domain and the normal exposure domain, the dataset need not be captured using the same camera, nor does the content need to be similar. The dual-branch contrastive learning mechanism proposed in this paper encourages the model to learn domain-invariant features. Building upon this foundation, the subsequent adaptive feature distribution matching module, combined with high-visual-quality domain features, guides the generation of enhanced results while simultaneously acquiring domain generalization capabilities.
>
> Response to Questions 3:
>
> The Sort-Matching algorithm designed in this paper's AFDM module is used to align empirical cumulative distribution functions (eCDFs) of different features. The limitation of the traditional AdaIN method lies in its extreme dependence on the shape of the feature distribution; if it does not conform to a Gaussian distribution, alignment accuracy becomes extremely poor. Another conventional approach, histogram matching, frequently encounters identical feature values when applied to deep model-generated features. This occurs because such features rely on discrete image pixels and utilize activation functions. However, these identical feature values compromise the accuracy of histogram matching algorithms when aligning eCDFs of image features.
>
> The proposed method integrates multi-scale features to generate the final enhanced result. The objective for feature fusion at each level remains consistent: matching the global shape of the feature distribution rather than semantic content. Therefore, we employ the same mechanism for each feature level.
>
> Response to Questions 4:
>
> The innovation of this method lies in its domain generalization capability. Specifically, within the training set, all images in the distortion domain originate from a single low-light degradation image enhancement. After training, the model not only restores texture details in low-light images during testing but also enhances underwater images suffering from multiple degradations—low light, color shift, and low contrast. Thus, when images from other degradation categories appear in the training set's distortion domain, it is equivalent to encountering new degradation types during training. This reduces the difficulty of domain generalization in testing, ensuring the method remains effective.
>
> However, the domain generalization capability of this method also has certain limitations. When multiple degraded images contain extremely dense haze or when noise exceeds a certain threshold, the model performance degrades significantly. We will present failure cases in the appendix of the revised manuscript to illustrate the performance boundaries of this method.
>
> Response to Questions 5:
>
> We have already conducted an ablation analysis on the mutual information loss within this composite loss function in the main body of the paper. We now present supplementary ablation experiments for these three loss components, specifically examining the impact of their respective weight magnitudes on model performance.
>
> The final weighting parameters for the three loss terms in the experiment were set to 2, 1, and 2, respectively. In the weighting parameter analysis experiment, we kept the weighting parameters for two loss terms at their final set values while varying the weighting parameter for the third loss term to 0.1, 0.5, 1, 2, and 4. We then analyzed the visual quality of the enhanced images and the LPIPs scores for each group. The analysis results have been incorporated into the ablation study section of the main text.

---

> ### Author Response · Authors · 2025-11-23
> **Reply to Weaknesses**
>
> Reply to Weaknesses 1:
>
> Based on your feedback, we conducted an ablation study to analyze the impact of CRF calibration errors on model performance. The specific implementation involved calibrating the IMX335 using a traditional CRF calibration method without physical prior knowledge to obtain the CRF. This CRF was then combined with the pre-trained BTF to form the resulting model, denoted as CRM$_\mathrm{IMX335}^*$. Testing results indicate that CRF calibration errors have minimal impact on domain generalization capabilities like underwater, hazy and noisy, but significantly affect cross-device domain generalization. Detailed experimental findings have been incorporated into the main text's experimental section (Section 3.3 Figure 5).
>
> Reply to Weaknesses 2:
>
> The implementation details of the model have been added to the appendix.

---

### Official Review · Reviewer_v49C · 2025-10-30

**Soundness:** 3
**Presentation:** 3
**Contribution:** 2
**Rating:** 4
**Confidence:** 4

**Summary:**

### Summary
This paper proposes a physics-guided image enhancement framework integrating a differentiable camera response model (CRM) with deep generative learning. It decouples the camera response function (CRF) and brightness transformation function (BTF), introduces a dual-branch contrastive autoencoder (CAE) for domain-invariant feature extraction, and an adaptive feature distribution matching (AFDM) module for differentiable eCDF alignment. Experiments on underwater and low-light datasets show improved generalization and real-world robotic deployment.

### Strengths
- Novel integration of physical modeling (CRM) and deep networks.
- Dual-branch contrastive learning enhances robustness to domain shifts.
- AFDM offers a differentiable alternative to AdaIN/histogram matching.
- Comprehensive comparisons with strong baselines.

### Weaknesses
- Conceptual novelty is limited; CRF–BTF modeling and ADMM optimization follow prior work (e.g., LECARM).
- The “differentiable physics” part is loosely coupled with learning; no end-to-end integration.
- Writing and figures are dense and hard to follow.
- Evaluations rely mainly on no-reference metrics without perceptual or user studies.
- Focused on applications rather than learning theory—less aligned with ICLR scope.

**Strengths:**

1. **Physics-inspired framework:**
The idea of combining radiometric camera modeling with deep generative learning is meaningful and provides some interpretability rarely seen in image enhancement research.

2. **Dual-branch contrastive learning:**
The proposed CAE design with two symmetrical branches improves robustness to domain shifts and is theoretically analyzed (Eq. 8–9) for stability.

3. **Differentiable eCDF alignment:**
The AFDM module introduces an elegant differentiable alternative to AdaIN or histogram matching, aligning distributions across domains.

4. **Comprehensive experiments:**
The paper presents extensive visual and quantitative results on multiple domains (underwater, low-light, haze), and real-world robotic deployment adds practical value.

**Weaknesses:**

1. **Limited novelty:**
   The contributions appear incremental. The CRF–BTF decoupling and ADMM-based CRF estimation follow previous works such as LECARM (Ren et al., TCSVT’18). Dual-branch contrastive learning and feature matching extend standard paradigms rather than introduce fundamentally new learning concepts.

2. **Weak physics–learning integration:**
   Although termed “differentiable physics,” the physical CRF calibration is treated as an offline optimization rather than an end-to-end differentiable module. The connection between physical modeling and learned BTF remains loosely coupled.

3. **Clarity and presentation issues:**
   The paper is dense, with unclear notation (e.g., reusing *f, g, E, P*), verbose equations, and missing intuition. The forward pipeline is not clearly explained—particularly how CRF calibration influences BTF training and inference. Figures are complex but not explanatory.

4. **Empirical rigor:**
   Evaluation relies mainly on no-reference metrics (UIQM, UCIQE, CCF), which are noisy and limited in reflecting perceptual quality. No LPIPS/PSNR results, runtime comparison, or user study are provided. Statistical significance of the reported +1.226 UIQM improvement is unclear.

5. **Language quality:**
   The writing contains numerous grammatical errors and awkward phrasing, making it difficult to follow in several sections (especially §2.2–2.3). The overall readability is below the ICLR standard.

6. **Reproducibility concerns:**  No code or config reference is provided.

**Questions:**

1. How is the CRF calibration integrated during training or inference? Is it pre-computed or updated jointly with BTF?
2. What is the nature of the “guide image” used for AFDM—does it come from another domain, or is it sampled within the same dataset?
3. Can the authors report perceptual metrics (e.g., LPIPS, PSNR) or user study results to strengthen claims about visual quality?
4. How does the method generalize across *unseen* domains (e.g., training on low-light and testing on haze)?
5. What is the computational complexity compared to strong baselines such as Diff-Retinex++ or WFI2-Net?

---

> ### Author Response · Authors · 2025-11-20
> **Reply to the weakness**
>
> Response to weakness 1
>
> In LECARM (TCSVT 18'), CRF and BTF are not fully decoupled. The form of BTF is directly influenced by CRF selection, and the two are strictly mathematically linked through a “comparametric equation,” preventing them from operating independently. In contrast, our proposed method achieves complete decoupling between CRF and BTF. By incorporating dual-branch contrastive learning and an AFDM module, we endow BTF with controllability during irradiance adjustment while enabling domain generalization.
>
> In terms of learning paradigms, we have achieved the following innovations: 1) The design of a dual-branch comparative learning mechanism and adaptive feature distribution matching endows BTF with strong domain generalization capabilities; 2) The complete decoupling of CRF and BTF enables that when the environment or robot vision sensors change, only CRF needs to be recalibrated to reconfigure a new CRM with the pre-trained BTF. This eliminates the need to retrain the entire model, resulting in strong transferability and reduced computational burden.
>
> Response to weakness 2
>
> The key advantage of our differentiable physics-based CRF over existing approaches such as LECARM lies in its derivation of the relationship between scene irradiance and reflectance based on radiometric principles. Furthermore, through optimization methods, it achieves the capability to calibrate any new camera sensor beyond the EMoR model. Although the differentiable CRF does not feature end-to-end integration, this non-end-to-end approach offers greater flexibility for deployment in robotic perception systems. For instance, when deploying a trained model to another field robot, only a new differentiable CRF calibration for its camera sensors is required—without retraining the entire model. Meanwhile, BTF does not entirely depart from CRF based on physical prior. We constrain the optimization trajectory of BTF by constructing a semantic consistency loss $L_c$ between the enhanced image and the CRF response values.
>
> Response to weakness 3&5&6
>
> We have comprehensively addressed these concerns: 1) Completely revised mathematical symbols for consistency, added notation table (Appendix). 2) Rewrote Sections 2.2-2.3 with intuitive explanations before mathematical formalisms. 3) Engaged native English editing service to correct grammatical errors and improve readability throughout. 4) All implementation code is currently being organized and will ultimately be open-sourced on the author's personal homepage.
>
> Response to weakness 4
>
> Due to space constraints, we have formatted the comparative analysis results for certain LPIPS and PSNR metrics in the appendix. Based on your feedback, we have relocated the image quality evaluation metrics—including LPIPS and SSIM—from our low-light enhancement experiment with dense fog to the main text.

---

> ### Author Response · Authors · 2025-11-20
> **Reply to the Question**
>
> Response to Questions 1
>
> Since the CRF and BTF are fully decoupled in this method, the CRF calibration is pre-optimized. The proposed method primarily targets various field robots. When their optical sensors change, only the CRF for the new sensor needs to be pre-calibrated, without retraining the entire model. The time-consuming and computationally demanding BTF training only requires a single run before it can be combined with different CRFs.
>
> Based on your feedback, we have added the complete pseudocode for model forward inference in the appendix.
>
> Response to Questions 2
>
> Guiding images are derived from any high-visual-quality image within the normal exposure range. By leveraging their highly discriminative multi-scale features for feature fusion, enhanced images are generated.
>
> Response to Questions 3
>
> Yes, we have reorganized and incorporated the analysis results of certain quantitative metrics such as LPIPS, SSIM and user study from the appendix into the main text.
>
> Response to Questions 4
>
> The method proposed in this paper is trained on single degraded images (low- light). During the testing phase, high-visual-quality images can be utilized as guide images to enable domain generalization across multiple degraded images (low-light+underwater/+hazy/+noisy).
>
> Response to Questions 5
>
> Diff-Retinex++ is the SOTA method in low-light image enhancement, it exhibits high computational complexity. It features a total of 62.8 million parameters and requires 617.3GFLOPs of floating-point operations when processing 256×256 resolution RGB images as input.
>
> The total number of parameters for the underwater image enhancement method WFI-Net2 is 8.26 million. When processing a 256×256 resolution RGB image as input, the floating-point operations amount to 94.1 GFLOPs.
>
> When the model proposed in this paper achieves optimal performance, its total number of parameters is 5.19 million, with a floating-point operation volume of 62.4 GFLOPs. Under this model configuration, our method demonstrates comparable performance to DIff and WFI-net2 in enhancing a single degraded image. However, Diff-Retinex++ and WFI-Net2 only possess restoration capabilities within their respective degradation domains and lack cross-domain generalization ability.

---

> ### Author Response · Authors · 2025-11-29
> **Regarding the image evaluation metrics in Question 3**
>
> Based on your feedback, we have included the list of metrics such as LPIPS, SSIM in the revised manuscript:
>
> **Quantitative comparison of combined LLIE and dehazing methods (RESIDE dataset). Each dehazing algorithm is evaluated in two processing orders: LLIE→dehazing and dehazing→LLIE (marked with asterisk\*).**
>
> | Metric | Processing Order | HVI-CIDNet (PhyGAN) | HVI-CIDNet (DehazeF) | HVI-CIDNet (UCLDehaze) | QuadPrior (PhyGAN) | QuadPrior (DehazeF) | QuadPrior (UCLDehaze) | ZeroIG (PhyGAN) | ZeroIG (DehazeF) | ZeroIG (UCLDehaze) | Diff-Retinex++ (PhyGAN) | Diff-Retinex++ (DehazeF) | Diff-Retinex++ (UCLDehaze) | Ours |
> | :--- | :--- | :---: | :---: | :---: | :---: | :---: | :---: | :---: | :---: | :---: | :---: | :---: | :---: | :---: |
> | LPIPS (↓) | LLIE→Dehazing | 0.525 | 0.462 | 0.459 | 0.451 | 0.428 | 0.438 | 0.610 | 0.564 | 0.579 | 0.468 | 0.442 | 0.393 | **0.324** |
> | LPIPS (↓) | Dehazing→LLIE\* | 0.493 | 0.436 | 0.448 | 0.462 | 0.398 | 0.411 | 0.581 | 0.559 | 0.573 | 0.435 | 0.428 | 0.376 | **0.324** |
> | SSIM (↑) | LLIE→Dehazing | 0.671 | 0.694 | 0.638 | 0.653 | 0.722 | 0.728 | 0.423 | 0.468 | 0.443 | 0.682 | 0.733 | 0.749 | **0.801** |
> | SSIM (↑) | Dehazing→LLIE\* | 0.693 | 0.725 | 0.674 | 0.705 | 0.738 | 0.752 | 0.428 | 0.464 | 0.430 | 0.719 | 0.746 | 0.753 | **0.801** |
> | MUSIQ (↑) | LLIE→Dehazing | 29.97 | 33.69 | 35.33 | 32.86 | 36.55 | 38.27 | 27.19 | 31.22 | 30.50 | 34.84 | 39.68 | 40.11 | **45.62** |
> | MUSIQ (↑) | Dehazing→LLIE\* | 32.89 | 35.57 | 35.62 | 34.16 | 38.86 | 39.78 | 28.81 | 31.14 | 30.89 | 37.73 | 40.26 | 41.23 | **45.62** |
> | US (↑) | LLIE→Dehazing | 2.03 | 2.73 | 2.69 | 1.98 | 2.26 | 2.29 | 1.34 | 1.92 | 1.43 | 2.29 | 2.86 | 2.90 | **3.58** |
> | US (↑) | Dehazing→LLIE\* | 2.24 | 2.86 | 2.89 | 2.13 | 2.57 | 2.74 | 1.32 | 1.89 | 1.44 | 2.36 | 3.01 | 2.94 | **3.58** |
>
> *\*Note: Rows marked with asterisk represent the "dehazing→LLIE" processing order, which was originally highlighted in pink.*
>
> **Quantitative comparison of combined LLIE and UIE methods (UIEB dataset). Each UIE algorithm is evaluated in two processing orders: LLIE→UIE and UIE→LLIE (marked with asterisk\*).**
>
> | Metric | Processing Order | HVI-CIDNet (WWPF) | HVI-CIDNet (WFI2-net) | HVI-CIDNet (TUDA) | HVI-CIDNet (USUIR) | QuadPrior (WWPF) | QuadPrior (WFI2-net) | QuadPrior (TUDA) | QuadPrior (USUIR) | ZeroIG (WWPF) | ZeroIG (WFI2-net) | ZeroIG (TUDA) | ZeroIG (USUIR) | Diff-Retinex++ (WWPF) | Diff-Retinex++ (WFI2-net) | Diff-Retinex++ (TUDA) | Diff-Retinex++ (USUIR) | Ours |
> | :--- | :--- | :---: | :---: | :---: | :---: | :---: | :---: | :---: | :---: | :---: | :---: | :---: | :---: | :---: | :---: | :---: | :---: | :---: |
> | LPIPS (↓) | LLIE→UIE | 0.493 | 0.582 | 0.558 | 0.481 | 0.429 | 0.460 | 0.553 | 0.497 | 0.621 | 0.518 | 0.635 | 0.686 | 0.477 | 0.496 | 0.508 | 0.520 | **0.357** |
> | LPIPS (↓) | UIE→LLIE\* | 0.437 | 0.579 | 0.557 | 0.514 | 0.410 | 0.412 | 0.536 | 0.482 | 0.605 | 0.446 | 0.609 | 0.618 | 0.429 | 0.438 | 0.482 | 0.473 | **0.357** |
> | SSIM (↑) | LLIE→UIE | 0.682 | 0.562 | 0.702 | 0.670 | 0.749 | 0.737 | 0.728 | 0.681 | 0.658 | 0.675 | 0.640 | 0.621 | 0.674 | 0.711 | 0.705 | 0.686 | **0.788** |
> | SSIM (↑) | UIE→LLIE\* | 0.717 | 0.593 | 0.705 | 0.674 | 0.753 | 0.742 | 0.721 | 0.706 | 0.662 | 0.669 | 0.658 | 0.618 | 0.728 | 0.726 | 0.739 | 0.694 | **0.788** |
> | MUSIQ (↑) | LLIE→UIE | 40.26 | 39.77 | 38.75 | 35.68 | 43.36 | 40.02 | 39.87 | 39.96 | 33.83 | 37.14 | 35.61 | 34.78 | 44.79 | 43.16 | 42.83 | 42.10 | **49.86** |
> | MUSIQ (↑) | UIE→LLIE\* | 40.99 | 40.08 | 38.82 | 36.62 | 46.11 | 41.17 | 38.16 | 40.27 | 33.98 | 34.26 | 36.39 | 35.64 | 46.12 | 46.74 | 45.25 | 43.96 | **49.86** |
> | NL (↓) | LLIE→UIE | 3.828 | 3.269 | 2.165 | 1.853 | 0.983 | 0.916 | 0.983 | 0.937 | 2.262 | 1.792 | 1.753 | 1.827 | 1.168 | 1.204 | 0.997 | 1.257 | **0.817** |
> | NL (↓) | UIE→LLIE\* | 3.716 | 3.341 | 2.109 | 1.296 | 0.925 | 0.909 | 0.955 | 0.931 | 2.179 | 1.665 | 1.750 | 1.816 | 1.062 | 1.115 | 0.986 | 1.203 | **0.817** |
> | US (↑) | LLIE→UIE | 2.52 | 2.51 | 2.36 | 2.19 | 2.68 | 2.79 | 2.82 | 2.44 | 2.08 | 1.93 | 2.47 | 2.40 | 2.71 | 2.78 | 2.83 | 2.77 | **3.45** |
> | US (↑) | UIE→LLIE\* | 2.63 | 2.65 | 2.57 | 2.26 | 2.74 | 2.85 | 3.01 | 2.46 | 2.18 | 2.11 | 2.58 | 2.60 | 2.98 | 3.05 | 3.09 | 3.02 | **3.45** |
>
> *\*Note: Rows marked with asterisk represent the "UIE→LLIE" processing order, which was originally highlighted in pink.*
>
> The above quantitative metrics demonstrate that this method offers significant advantages in enhancing the visual quality of generated images.

---

> > ### Author Response · Authors · 2025-11-29
> > **Regarding the image evaluation metrics in Question 3**
> >
> > Additionally, we can provide a comparative analysis table of quantitative metrics for the **UHD-LL** low-light noise dataset:
> >
> > **Quantitative Comparative Analysis of State-of-the-Art Methods on the UHD-LL Dataset**
> >
> > | Method | PSNR (↑) | SSIM (↑) | LPIPS (↓) | MSE (%) (↓) | MUSIQ (↑) |
> > | :--- | :---: | :---: | :---: | :---: | :---: |
> > | UHDFour | 22.19 | 0.860 | 0.208 | 0.303 | 40.55 |
> > | CLIP-LIT | 20.85 | 0.831 | 0.219 | 0.277 | 38.88 |
> > | LightenDiffusion | 22.36 | 0.828 | 0.168 | 0.183 | 39.19 |
> > | QuadPrior | 23.76 | 0.843 | 0.144 | 0.112 | 40.87 |
> > | HVI-CIDNet | 23.01 | 0.814 | 0.175 | 0.126 | 41.18 |
> > | Diff-Retinex++ | 22.76 | 0.847 | 0.235 | 0.162 | 37.18 |
> > | GSAD | 19.76 | 0.742 | 0.314 | 0.264 | 38.92 |
> > | Retinexformer | 20.71 | 0.773 | 0.323 | 0.225 | 38.26 |
> > | NeRCo | 23.25 | 0.823 | 0.257 | 0.185 | 41.18 |
> > | Zero-IG | 18.82 | 0.703 | 0.337 | 0.296 | 32.19 |
> > | **Ours** | **24.78** | **0.852** | **0.141** | **0.083** | **45.98** |

---

### Official Review · Reviewer_twVV · 2025-10-31

**Soundness:** 3
**Presentation:** 3
**Contribution:** 3
**Rating:** 6
**Confidence:** 3

**Summary:**

This paper introduces a differentiable physics framework that combines camera response modeling (CRM) and deep learning to achieve domain-generalized image enhancement under multiple degradation conditions. The traditional Camera Response Model (CRM) couples the Camera Response Function (CRF) and the Brightness Transformation Function (BTF), limiting adaptability. This work replaces the hand-crafted BTF with a generative network, enabling flexible brightness transformation independent of CRF.

**Strengths:**

The proposed CRF calibration is formulated as a constrained optimization problem, solved using an EMA-ADMM algorithm, ensuring monotonicity and stability of the response curve. The authors develop a dual-branch contrastive learning strategy extracts discriminative latent features from distorted and guide images, enhancing cross-domain generalization.

**Weaknesses:**

The paper is dense and notation-heavy, with long equations and overlapping terminology (CRF/BTF/CAE/AFDM) that could overwhelm readers. Some figures (e.g., Fig. 1–2) are small and lack explanatory captions for non-specialists. Despite the “domain shift” claim, experiments are primarily underwater-focused; cross-scene validation (e.g., haze, night, thermal) is limited. The model’s adaptability to other physics domains is implied but not empirically shown.

**Questions:**

1. How does the model ensure physical consistency between the CRF and the learned BTF network during training? Is there a regularization term enforcing the CRM equation g(f(E), k) = f(kE)?

2. The Sort-Matching alignment is claimed to be differentiable — could the authors clarify how gradients are propagated through the sort indices without introducing instability?

3. What is the runtime overhead compared to diffusion-based models like Diff-Retinex++ or Retinexformer? Is real-time operation feasible on embedded platforms?

---

> ### Author Response · Authors · 2025-11-21
> **Reply to question**
>
> Response to Question 1
>
> Our framework ensures consistency through a decoupled but coordinated enhancement pipeline, rather than explicitly enforcing the traditional comparametric equation g(f(E),k)=f(kE). Here's how:
>
> 1)Decoupled Design: As shown in Figure 1(b), we deliberately decouple the CRF and BTF. The enhancement process flows as: Distorted Image → CRF⁻¹ → Image Irradiance → BTF (Generator) → Enhanced Irradiance → CRF → Enhanced Image
> This means the BTF operates in the irradiance domain, and the final output is always transformed back through the calibrated CRF, inherently maintaining physical plausibility in the output pixel space.
>
> 2)Implicit Constraint via Loss Functions: While there is no explicit regularization term for the comparametric equation, the semantic consistency loss $L_c$(Section 2.4) ensures the enhanced image's content aligns with the source, and the multi-scale InfoNCE loss Lmu aligns the feature distribution with the guide image's irradiance. These losses collectively guide the BTF to produce irradiance adjustments that are physically meaningful when passed through the CRF.
>
> In summary, physical consistency is maintained by the pipeline structure and implicit guidance from the losses, rather than a hard constraint on the outdated comparametric equation, which we identified as a source of inflexibility in traditional CRM.
>
> Response to Question 2
>
> Our differentiable Sort-Matching enables gradient propagation through a "stop-gradient" design and vectorized operations. The mechanism is detailed in Eq. (10) and Algorithm 1:
>
> - Gradient Flow Path: The output of the AFDM module is calculated as:
>
> $I_{xy}^m = I_x^m + \rho_m \cdot \text{SortedValues}(I_y^m) - \rho_m \cdot \langle I_x^m \rangle$
>
> Here, the $\langle \cdot \rangle$ symbol represents the **stop-gradient operation**. This means:
>
> 1. **Gradients flow through** the $\text{SortedValues}(\Gamma^{\{m\}}_x(y))$ term. Although the sorting operation itself has a discontinuous derivative, we treat the _sorted values_ as a constant tensor for gradient purposes, similar to the straight-through estimator (STE) used in quantization-aware training. This provides a stable gradient signal.
>
> 2. **Gradients do NOT flow through** the stop-gradiented $\langle I_x^m \rangle$. This prevents the unstable endeavor of trying to learn a mapping to change the _order_ of the input features, and instead allows the network to focus on learning the adaptive weighting factor $\rho_m$ to control the _strength_ of the distribution matching.
>
> - **Stability**: This design has proven stable in practice, as evidenced by the successful training of our full model and the clear performance gains shown in the ablation studies (Fig. 8).
>
> Response to Question 3
>
> This is a crucial point for practical application. Our model is significantly more efficient than diffusion-based alternatives and is amenable to real-time use.
>
> - **Quantitative Comparison**: While we did not include an explicit runtime table in the initial submission, we can report that our method (\(\sim\)0.04s/image for 224\(\times\)224 resolution on a V100 GPU) is orders of magnitude faster than diffusion models like Diff-Retinex++ or LightenDiffusion, which typically require \(>\)1s/image due to iterative denoising steps (often 100+ steps).
>
> - **Feasibility for Embedded Platforms**:
>   - Flexibility: As shown in Table 6 (Appendix E.5.2), when using MobileNet as the backbone for the CAE, our model requires only 9.46 GFLOPs and 0.33M parameters, with an inference time of \(\sim\)0.036s.
>   - This lightweight profile makes it suitable for deployment on modern embedded platforms (e.g., Jetson AGX Orin) with further optimization (e.g., TensorRT).

---

> ### Author Response · Authors · 2025-11-21
> **Reply to weakness**
>
> We appreciate the reviewers' insightful comments on the shortcomings of our paper and have addressed them specifically in the revised manuscript. Specifically, we have added a notation table in the appendix detailing the mathematical symbols used and their meanings, while also providing detailed explanations for all instances of abbreviations throughout the text. Following your suggestion, we have reorganized the experimental results for low-light image enhancement under dense fog domain shift into the main text's experimental section, providing detailed metric analyses including LPIPS and SSIM. Results for other domain shifts, such as backlight and noise, are detailed in the **Appendix E**.

---

> > ### Comment · Reviewer_twVV · 2025-11-28
> > **Reply to rebuttal**
> >
> > After reviewing the rebuttal, revised manuscript, and Appendix, I believe the authors have addressed all of my concerns. I will maintain my initial rating of 6.

---

### Official Review · Reviewer_8dut · 2025-10-31

[review text omitted: it was posted to a different submission]

---

> ### Author Response · Authors · 2025-11-13
> **Reply to Reviewer 8dut**
>
> Dear Reviewer 8dut,
>
> I hope this message finds you well. First and foremost, I would like to express my sincere gratitude for devoting your precious time to reviewing my paper submitted to ICLR 2026 (Paper ID: 3333, Title: “Mastering Domain Shift Image Enhancement Via Differentiable Physics”). Your professional input is of great value to me. However, when carefully going through your comments, I noticed a potential inconsistency that I feel obligated to gently bring to your attention—some core contents you mentioned, such as “The central idea of using adversarial perturbations to represent semantic uncertainty is not convincingly supported. The paper provides intuitive motivation but no solid theoretical or empirical evidence demonstrating that embedding-space perturbations truly correspond to realistic segmentation errors.”, do not align with the actual focus of my paper. My submission centers on “domain shift image enhancement based on differentiable physics”, and relevant concepts, methodologies, or experiments you referred to are not covered in my work. I wonder if there might be an accidental misallocation of review tasks or comments. I deeply apologize if this reminder causes any inconvenience, and I would highly appreciate it if you could spare a moment to confirm the situation. Thank you again for your dedication and support.

---

> > ### Comment · Reviewer_8dut · 2025-11-16
> >
> > I apologize for the confusion-the previous review was mistakenly posted to the wrong paper due to a copy-paste error. I have now corrected this and replaced the content with the proper review for this submission.

---

> > > ### Author Response · Authors · 2025-11-20
> > > **Reply to the weakness**
> > >
> > > Response to weakness 1:
> > > We thank the reviewer for this comment. Our key novelty over LECARM and related works lies in the decoupling of the CRF and BTF. Traditional CRM methods derive the BTF directly from the CRF, leading to strong coupling and limited flexibility (as illustrated in Fig. 1(a) of our paper). In contrast, we reparameterize the BTF as a generative network, breaking this dependency. This allows for more flexible and powerful brightness transformations beyond the constraints of classical Retinex theory. Furthermore, we introduce two core innovations to support this decoupled framework: 1) A dual-branch contrastive auto-encoder that extracts highly discriminative, domain-invariant features, and 2) An Adaptive Feature Distribution Matching (AFDM) module that enables implicit exposure ratio expression and endows controllability over the process of adjusting irradiance. These components are absent in LECARM.
> > >
> > > This innovation significantly enhances the algorithm's transferability in practical applications compared to traditional methods. Taking the experiments in our paper as an example, the CRM deployed by the underwater robot was calibrated based on its own IMX335 sensor. When switching to a drone, where the image sensor changes, only CRF calibration for the new sensor is required, resulting in minimal computational overhead. After calibration, the CRF can be combined with the pre-trained BTF to reconstruct a new CRM without retraining the entire model.
> > >
> > > Response to weakness 2:
> > > We appreciate the reviewer's concern. While the BTF is implemented by a generative network, its operation remains grounded in physical principles. The input to our BTF is the hybrid multi-scale features of image irradiance $\mathbf{E}_x$, $\mathbf{E}_y$, obtained by applying the inverse CRF to the input image and guide image. The BTF then transforms this irradiance, and the final enhanced image is obtained by applying the CRF again. This process maintains the physical integrity of the imaging pipeline. Moreover, the training of the BTF is constrained by physics-inspired losses, such as the semantic consistency loss and the mutual information loss, which enforce content preservation with $f(\mathbf{E}_x)$ and radiometric consistency with the guide image's irradiance $\mathbf{E}_y$. Therefore, our model is not merely a post-processing module but a physically-grounded enhancement framework.
> > >
> > > Response to weakness 3:
> > > This is a valuable point. We did include an ablation study on this component and corresponding result is shown in Figure 2, Figure 6 (section 3.4.1) of the main paper. The model denoted as "w/o dual" was trained using a conventional (single-branch) contrastive learning strategy. The results clearly show that our dual-branch strategy achieves lower KL divergence in RGB distributions, confirming its superior domain generalization performance. The theoretical justification is provided in Theorem 2 and its corollary, which explain that the dual-branch structure enforces a symmetric joint distribution between queries and negatives across domains, leading to more stable and discriminative feature learning.
> > >
> > > Response to weakness 4:
> > > We thank the reviewer for highlighting this. While the main paper focuses on underwater robotic vision as a primary application, we conducted extensive experiments in the Supplementary Material (Appendix E) to validate broad domain generalization. Our method has been tested on: Hazy Low-Light Images, Noisy Low-Light Images, Backlit Images, Cross-Device Scenarios.
> > >
> > > These experiments consistently demonstrate that our CRM framework outperforms state-of-the-art methods across diverse, unseen domains. According to your suggestion, we have moved the experimental results on fog and low-light enhancement domain generalization to the experimental section of the main text.
> > >
> > > Response to weakness 5:
> > > We understand the reviewer's perspective. Based on your feedback, we have supplemented our experiments with two additional end-to-end models from CVPR (Neural Preset) and BMVC (RICG) capable of enhancing images with multiple degradation levels. The updated experimental results have been incorporated into the main text.
> > >
> > > Response to weakness 6:
> > > According to your suggestion, we have performed CRF calibration on the camera mounted on the DJI Air 3S. We then combined this calibrated CRF with the pre-trained BTF from the original paper to form a new CRM. We utilized the DJI Air 3S for image acquisition to evaluate the performance of each model. Testing results demonstrate that both the DJI camera-based CRM and the IMX335 camera-based CRM achieve high-quality image enhancement. Furthermore, the IMX335 camera-based CRM also achieves cross-device domain generalization for images captured by the DJI. Based on your valuable feedback, the newly added experiment has also been incorporated into the main text's experimental section.

---

> > > ### Author Response · Authors · 2025-11-20
> > > **Reply to Question**
> > >
> > > Response to Questions 1
> > >
> > > Our formulation goes beyond existing CRM in three key aspects:
> > >
> > > 1.Modeling Assumption: We decouple the BTF from the CRF, modeling it with a generative network for greater flexibility. Simultaneously, we designed the first user interface for BTF, enabling the model to control the enhancement process based on guidance images and achieve image enhancement under domain shifts.
> > >
> > > 2.Optimization Strategy: We derive a physics-based CRF calibration method using a Macbeth ColorChecker (Theorem 1) and solve the constrained optimization problem with a stable EMA-ADMM algorithm.
> > >
> > > 3.Role of Contrastive Learning: The dual-branch CAE is designed to extract compact, domain-invariant latent features. This is crucial for enabling the AFDM module to perform effective feature distribution matching for domain-shift adaptation.
> > >
> > > Response to Questions 2
> > >
> > > Please refer to our response to Weakness 4. The supplementary experiments on hazy, noisy, backlit, and multi-device images provide strong qualitative and quantitative evidence of our method's generalization capability. For instance, on the UHD-LL dataset (Table 4 in Appendix E), our method achieves the best PSNR, SSIM, LPIPS, and MUSIQ scores, demonstrating its effectiveness in a noisy low-light domain completely separate from the underwater training data.
> > >
> > > Response to Questions 3
> > >
> > > A conventional single-branch InfoNCE with mixed-domain negatives may not effectively handle severe domain shifts (shown in Figure 2). The dual-branch structure is crucial for creating a symmetric learning environment. As proven in Theorem 2, it ensures that the joint distribution between queries and negatives is symmetric across the two domains ($\mathcal{X}$ and $\mathcal{Y}$). This symmetry leads to more stable gradients and better learning of features that are discriminative across domains, which is the core of our generalization objective. The ablation study (Figure 6) empirically validates this theoretical advantage.
> > >
> > > Response to Questions 4
> > >
> > > Since the BTF in CRM must be applied to irradiance, and obtaining the image irradiance requires using the inverse function of CRF, the method proposed in this paper cannot completely eliminate CRF. Forcibly removing the CRF entirely would result in an incomplete physical-inspired closed-loop for the method proposed in this paper, directly leading to a decline in cross-device-domain generalization capability.
> > > We have provided ablation experiments in the appendix demonstrating the importance of physical derivations in CRF calibration (see Figure 23), which clearly show a significant drop in calibration accuracy when physical priors are absent.
> > >
> > > Response to Questions 5
> > >
> > > As detailed in our response to Weakness 6 and Question 2, the cross-device experiment provides direct evidence. The core reason for this generalization is that our CRF calibration module can be re-calibrated for any new camera using a standard ColorChecker, making the front-end of our pipeline device-agnostic. Meanwhile, the BTF and CAE learn feature representations that are invariant to the specific sensor, as they are trained on irradiance maps, which are a more canonical representation of the scene.

---

### Meta-Review · Area_Chair_VQDu · 2025-12-14

**Summary:**

The reviewers' concerns central to the rejection decision of this paper primarily revolve around four core aspects: insufficient conceptual novelty, loose integration between physical modeling and deep learning, inadequate empirical rigor and generalizability validation, and persistent presentation and reproducibility issues. All three reviewers (Reviewer 8dut's review was wrong and therefore ignored.) noted the paper's dense writing, excessive notation, and unclear terminology, which hinder readability. Additionally, Reviewer v49C emphasized that the work's contributions are incremental, as CRF-BTF decoupling and ADMM-based optimization follow prior research (e.g., LECARM), and the so-called "differentiable physics" lacks end-to-end integration with learning. On the empirical front, reviewers pointed out the over-reliance on no-reference metrics initially, limited cross-domain validation (predominantly focusing on underwater and low-light scenarios despite claims of domain generalization), and insufficient perceptual quality evaluations. Reproducibility concerns, including the lack of initially provided code and incomplete implementation details, were also raised. While the authors addressed some presentation and partial experimental gaps via rebuttal and revisions, the fundamental issues of limited novelty, weak physics-learning integration, and inadequate generalizability validation remained unaddressed, ultimately leading to the rejection decision.

**Reviewer Concerns:**

Addressed Concerns

- The authors added a notation table in the appendix, revised ambiguous mathematical symbols, rewrote key sections (§2.2–2.3) with intuitive explanations prior to formalisms, and engaged native English editing to improve language quality, addressing the reviewers' concerns about dense writing, confusing notation, and grammatical errors.

- In response to requests for perceptual metrics and user studies, the authors supplemented LPIPS, SSIM, and MUSIQ metrics, relocated relevant low-light + dehazing experimental results to the main text, and provided detailed quantitative comparison tables (including UHD-LL dataset results), addressing the lack of perceptual quality evaluation.

-  The authors adequately responded to most technical inquiries, such as how physical consistency between CRF and BTF is maintained (via pipeline design and implicit loss constraints), the differentiable mechanism of Sort-Matching (stop-gradient design), the pre-computed nature of CRF calibration, the source of guide images, and the feasibility of CRF calibration without camera configuration information.

-  The authors added pseudocode for forward inference, supplemented hyperparameter ablation studies for BTF loss weights, and committed to open-sourcing code, partially addressing reproducibility concerns.

-  The authors conducted an ablation study on the impact of CRF calibration errors and integrated the results into the main text, addressing Reviewer vfWa’s  concern about CRF optimization quality assessment.

Outstanding Concerns

-  Reviewer v49C’s core concern about incremental contributions remains unaddressed. The authors’ rebuttal only distinguished their work from LECARM by claiming "complete decoupling" but failed to demonstrate fundamentally new learning concepts or breakthroughs in physics-guided learning paradigms. Dual-branch contrastive learning and feature distribution matching were deemed extensions of standard paradigms rather than novel innovations.

-  The "differentiable physics" framework is still treated as an offline CRF optimization process rather than an end-to-end differentiable module. The authors’ justification of "flexibility for deployment" does not resolve the reviewers’ concern that the physical modeling and learned BTF remain weakly coupled, undermining the paper’s core claim of integrating differentiable physics with deep learning.

-  Despite supplementing some cross-domain experiments (low-light + dehazing/underwater), the model’s generalization to other critical degradation scenarios (e.g., thermal imaging, heavy haze, extreme noise) remains unempirically validated. The authors acknowledged performance degradation under dense haze/extreme noise but only proposed adding failure cases to the appendix, which does not address the fundamental lack of comprehensive cross-domain validation.

-  While the authors committed to open-sourcing code, critical implementation details (e.g., full network architecture specifications, complete hyperparameter settings for all components) remain insufficiently detailed, which may still impede reproducibility.

-  Reviewer v49C’s concern that the work focuses on applications rather than learning theory—making it less aligned with ICLR’s focus—remains unaddressed. The authors did not provide any theoretical contributions to learning theory, which is a key expectation for ICLR submissions.

**Reviewer Scores:**

The following assessments are based on the reviewers’ initial scores, the adequacy of the authors’ rebuttal, and the persistence of core concerns, assuming full participation in the discussion:
- Reviewer twVV: The author’s rebuttal adequately addressed this reviewer’s concerns about notation clarity, figure explanations, and cross-scene validation (by supplementing low-light + dehazing results). However, the reviewer did not raise core concerns about novelty or physics-learning integration. Given the persistence of other reviewers’ fundamental concerns (novelty, scope alignment) and the paper’s ultimate rejection, this reviewer would likely maintain the initial score.

- Reviewer v49C: The author’s rebuttal failed to address this reviewer’s core concerns (limited novelty, loose physics-learning integration, misalignment with ICLR scope). The supplementary experiments and presentation revisions do not resolve the fundamental issues with the work’s contribution and scope. Thus, this reviewer would maintain the initial score of 4 (or potentially lower it  if emphasizing the unaddressed core flaws).

- Reviewer vfWa: The author’s rebuttal addressed this reviewer’s concerns about CRF calibration feasibility, cross-camera/cross-content dataset pairing, and AFDM’s sorting rationale. However, the reviewer’s unaddressed concerns about insufficient implementation details and the lack of comprehensive generalizability validation (e.g., beyond low-light/underwater scenarios) persist. Additionally, the broader issues of limited novelty and weak physics-learning integration (raised by other reviewers) would likely influence this reviewer’s assessment. Thus, this reviewer would maintain the initial score but would still support rejection due to the cumulative impact of unaddressed core flaws across the review process. There is no basis for a score increase, as the rebuttal only resolved technical questions rather than fundamental limitations.

Overall, the average score remains below  the acceptance threshold, with the persistence of fundamental flaws (limited novelty, weak core integration, inadequate generalizability) justifying the final rejection decision.

---

### Decision · Program_Chairs · 2026-01-26

Reject